# COORDINATION SCHEME PROBING FOR GENERALIZABLE MULTI-AGENT REINFORCEMENT LEARNING

## ABSTRACT

Coordinating with previously unknown teammates without joint learning is a crucial need for real-world multi-agent applications, such as human-AI interaction. An active research topic on this problem is ad hoc teamwork, which improves agents' coordination ability in zero-shot settings. However, previous works can only solve the problem of a single agent's coordination with different teams, which is not in line with arbitrary group-to-group coordination in complex multi-agent scenarios. Moreover, they commonly suffer from limited adaptation ability within an episode in a zero-shot setting. To address these problems, we introduce the Coordination Scheme Probing (CSP) approach that applies a disentangled scheme probing module to represent and classify the newly arrived teammates beforehand with limited pre-collected episodic data and makes multi-agent control accordingly. To achieve generalization, CSP learns a meta-policy with multiple sub-policies that follow distinguished coordination schemes in an end-to-end fashion and automatically reuses it to coordinate with unseen teammates. Empirically, we show that the proposed method achieves remarkable performance compared to existing ad hoc teamwork and policy generalization methods in various multi-agent cooperative scenarios.

## 1 INTRODUCTION

Multi-Agent Reinforcement Learning (MARL) (Gronauer & Diepold, 2021) holds promise in numerous cooperative domains, including resource management (Xi et al., 2018), traffic signal control (Du et al., 2021), and autonomous vehicles (Zhou et al., 2020). A number of methods (Lowe et al., 2017; Wang et al., 2020b; Papoudakis et al., 2019; Christianos et al., 2021) have been proposed to deal with joint policy learning and scalability issues. However, previous works commonly assume a fixed team composition, where agents only need to coordinate with training partners and do not consider generalization. Such a process is not in line with real-world applications that require agents to cooperate with unknown teammates whose coordination schemes may not be explicitly available. When coordinating with distributed teammates, co-trained agents may fail (Gu et al., 2022).

Another research domain dedicated to this need is ad hoc teamwork (Stone et al., 2010), which stands at a single agent's perspective to adapt to different teams in a zero-shot fashion. However, current methods exist three significant limitations: (1) The differences between teammates could be very subtle and lie in only a few critical decisions. In this case, zero-shot coordination is not always feasible. As shown in the example in Fig. 1, teammate's behaviors are indistinguishable before the final critical action. (2) The information for identifying different teams is collected by the same policy that aims to maximize coordination performance. Thus, the exploration-exploitation dilemma will cause adverse effects on both sides. (3) Ad hoc teamwork stands at a single agent's perspective, which cannot deal with arbitrary group-to-group generalizations in complex multi-agent scenarios.

To overcome these limits and achieve generalizable coordination, we propose a multi-agent learning framework called Coordination Scheme Probing (CSP). Instead of doing zero-shot coordination, CSP tries to capture the unknown teammates' coordination scheme beforehand with limited pre-collected episodic data. Concretely, we start with generating a set of teams with high performance and diversified behaviors to discover different solutions to the task. After that, the scheme probing module learns to interact with these teams to reveal their behaviors and represents their policies with dynamics reconstruction (Hospedales et al., 2021) at the team level. Finally, we discover coordination

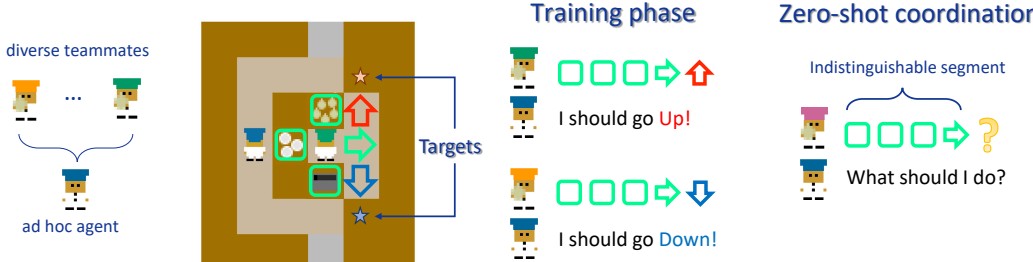

Figure 1: An example from Overcooked (Carroll et al., 2019a), where the chef in the blue hat needs to cooperate with chefs in other colors. Different chefs put the dishes in different positions after they are done cooking. For example, during training, the chefs in the green and orange hats put their dishes in the red and blue pentagrams as their last action, respectively. When a chef in the pink hat comes, how should the chef in the blue hat work with him? Since the first four actions for different chefs are indistinguishable, how to detect his type and coordinate with him is challenging.

schemes as clusters of the representations and train a multimodal meta-policy to adapt to them, with each sub-policy dedicated to a unique coordination scheme.

The whole learning framework of CSP learns multiple coordination schemes from a given task in an end-to-end fashion and automatically reuses the most suitable scheme recognized by the scheme probing module to coordinate with unknown partners. To validate our method, we conducted experiments on four challenging multi-agent cooperative scenarios and found that CSP achieves a more effective and robust generalization to unknown teammates compared to various baselines. Our visual analysis further confirms that CSP acquires multiple coordination schemes that are clearly distinguishable and can be precisely recognized. Meanwhile, our ablations show the necessity of probing unknown teammates in advance as well as using a multimodal policy for adaptation.

## 2 RELATED WORK

**Multi-Agent Reinforcement Learning (MARL).** Fully cooperative MARL methods (Foerster et al., 2018) mainly focus on deploying a fixed team with centralized training and decentralized execution (CTDE) setting (Oliehoek et al., 2008). Value factorization methods (Sunehag et al., 2018; Rashid et al., 2018; Wang et al., 2020a) are adopted widely in the CTDE process to solve the instability of teammates in the training process. Our work also utilizes the CTDE setting to learn policies collaborating with others. However, such a learning process may converge to a fixed modal and may not coordinate well with unseen teammates trained elsewhere. To deal with the issue, Tang et al. (2021) propose a method that could find a hard-to-search optimal policy by reward randomization. In game-theoretic methods, PSRO (Lanctot et al., 2017) aims at tackling the overfitting problem by learning meta-strategies, and $\alpha$-PSRO (Muller et al., 2020) further utilizes $\alpha$-Rank as an alternative for Nash solver to avoid equilibrium selection issues and improve efficiency. In this work, we borrow the idea of breaking local optima to mine diverse coordination behaviors from the environment, instead of training a single optimal policy.

**Ad Hoc Teamwork.** Ad hoc teamwork aims to learn a single autonomous agent that cooperates well with other agents without pre-coordination (Stone et al., 2010; Mirsky et al., 2022). Their methods include population based training (Strouse et al., 2021; Jaderberg et al., 2017; Carroll et al., 2019a; Charakorn et al., 2021), Bayesian belief update (Barrett et al., 2017; Albrecht et al., 2016), experience recognition (Barrett et al., 2017; Grover et al., 2018b; Chen et al., 2020), and planning (Bowling & McCracken, 2005; Ravula et al., 2019). A recent work ODITS (Gu et al., 2022) introduces an information based regularizer to automatically approximate the hidden of the global encoder with the local encoder. However, they are limited to learn a single ad hoc agent and only makes zero-shot adaptation. We take a step further to develop a framework for more complex team-to-team coordination under a few-shot setting.

**Policy Representation.** It is a well studied topic in multi-agent scenarios to anticipating others' behaviors and weaken the non-stationary issue (Albrecht & Stone, 2018). DRON (He et al., 2016)

uses a modeling network to reconstruct the actions of opponents from full history. Grover et al. (2018b) adopt the concept of interaction graph (Grover et al., 2018a) to learn a contrastive style representation with prepared policy class. DPIQN (Hong et al., 2018) learns "policy features" of other agents and incorporates them into the Q-network for better value estimation. LIAM (Papoudakis et al., 2021a) aims at estimating teammates' current actions and observations based on learning agent's local history alone. Inspired by dynamics-reconstruction (Raileanu et al., 2020) in meta-learning, we use an architecture that directly reconstructs the policy-dynamics. of a team policy, instead of predicting their temporal behaviors, to get a compact and precise representation.

## 3 BACKGROUND AND PROBLEM FORMALIZATION

**Cooperative Task.** We formalize the problem as a Dec-POMDP (Oliehoek & Amato, 2016) with a controllable group, defined as $\mathcal{M} = \langle \mathcal{N}, \mathcal{S}, \{\mathcal{O}_i\}_{i=1}^n, \{\mathcal{A}_i\}_{i=1}^n, \Omega, P, R, \gamma, d_0, G^1 \rangle$, where $\mathcal{N} = \{1, \ldots, n\}$ is the set of agents, $\mathcal{S}$ is the set of global states, $\mathcal{O}_i$ is agent $i$'s observation space, $\mathcal{A}_i$ is agent $i$'s action space, $\gamma \in [0, 1)$ is the discount factor, and $d_0$ denotes the initial state distribution. At each timestep, agent $i \in \mathcal{N}$ acquires a local observation $o_i \in \mathcal{O}_i$ with observation function $\Omega(s, i)$ and chooses an action $a_i \in \mathcal{A}_i$ via individual policy $\pi_i(a_i | \tau_i)$, where $\tau_i$ denotes input history. The joint action $\boldsymbol{a} = \langle a_1, \ldots, a_n \rangle$ transitions the system to next global state $s'$ according to the transition function $P(s' \mid s, \boldsymbol{a})$, and all agents get a shared reward $r = R(s, \boldsymbol{a})$. The group describes control range at test time, where $G^1 \subseteq \mathcal{N}$ is the subset of controllable agents, and its complementary $G^{-1}$ contains uncontrollable teammates that $G^1$ should adapt to. We denote join observation, action and policy for $G^1$ as $\boldsymbol{o}^1 = \langle o_i \rangle, \boldsymbol{a}^1 = \langle a_i \rangle, \boldsymbol{\pi}^1 = \langle \pi_i \rangle, i \in G^1$, and the corresponding parts for $G^{-1}$ are defined similarly. Thus, the (global) joint policy can be written as $\boldsymbol{\pi} = \langle \boldsymbol{\pi}^1, \boldsymbol{\pi}^{-1} \rangle$.

**Coordination Scheme.** We define this term to better describe generalization. Let $\Pi_f$ be the set of all joint policies with high coordination performance. Coordination scheme $C = \{c_i\}$ is defined as a partition of $\Pi_f$. Each coordination scheme $c_i$ is a set of joint policies, where $c_i \cap c_j = \emptyset$, if $i \neq j$ and $\bigcup_{c_i \in C} c_i = \Pi_f$. We assume that the coordination performance can be guaranteed if all the agents are in the same coordination scheme, even if they have minor differences. Otherwise, no such guarantee exists generally. Intuitively, $C$ is determined by the coordination task itself, different elements in which reflect different unique high-level joint behaviors. We may use words like "follows" or "is under" in this paper to describe the same thing as $\boldsymbol{\pi} \in c$ for ease in expression.

**Problem Formalization.** Our aim is to control $G^1$ to coordinate with $G^{-1}$ under any coordination scheme $c \in C$. We assume that $G^{-1}$ has no adaptation ability and will stick to a fixed scheme no matter what $G^1$ behaves. With a little abuse of notations, we use $\boldsymbol{\pi}^{-1} \in c$ to denote $\boldsymbol{\pi} \in c$ and $\boldsymbol{\pi}^{-1}$ is its slice for $G^{-1}$. Formally, the optimal policy $\boldsymbol{\pi}_\theta^{1*}$ parameterized by $\theta$ for $G^1$ is to maximize the discounted cumulative reward: $\theta^* = \arg\max_\theta \mathbb{E}_{\boldsymbol{\pi}^{-1} \in c, c \in C} \left[ \sum_{t=0}^{H-1} \gamma^t r_{t+1} \,\middle|\, (\boldsymbol{a}^1, \boldsymbol{a}^{-1}) \sim (\boldsymbol{\pi}_\theta^1, \boldsymbol{\pi}^{-1}), P, d_0 \right]$, where $r_{t+1}$ is the shared reward at timestep $t$ and $H$ is the episode length. Since we do not have direct access to the true scheme set $C$, we create a set of diverse policies $\Pi_{\text{train}}$ and directly sample $\boldsymbol{\pi}^{-1}$ from it as a surrogate of the two-factor sampling, and rewrite the objective as: $\theta^* = \arg\max_\theta \mathbb{E}_{\boldsymbol{\pi}^{-1} \in \Pi_{\text{train}}} \left[ \sum_{t=0}^{H-1} \gamma^t r_{t+1} \,\middle|\, (\boldsymbol{a}^1, \boldsymbol{a}^{-1}) \sim (\boldsymbol{\pi}_\theta^1, \boldsymbol{\pi}^{-1}), P, d_0 \right]$. Another diverse set $\Pi_{\text{eval}}$ is created to evaluate the generalization performance.

## 4 METHOD

This section describes how the CSP framework addresses the generalizable coordination problem in an end-to-end manner (Fig. 2). When given a cooperative task, CSP learns with three stages: (1) It generates a diverse population of team policies to discover multiple feasible coordination schemes. (2) It trains a scheme probing module to efficiently represent different teams by self-supervised team-dynamics reconstruction. (3) It discovers the underlying coordination schemes by clustering the representations and trains a multimodal meta-policy to adapt to them.

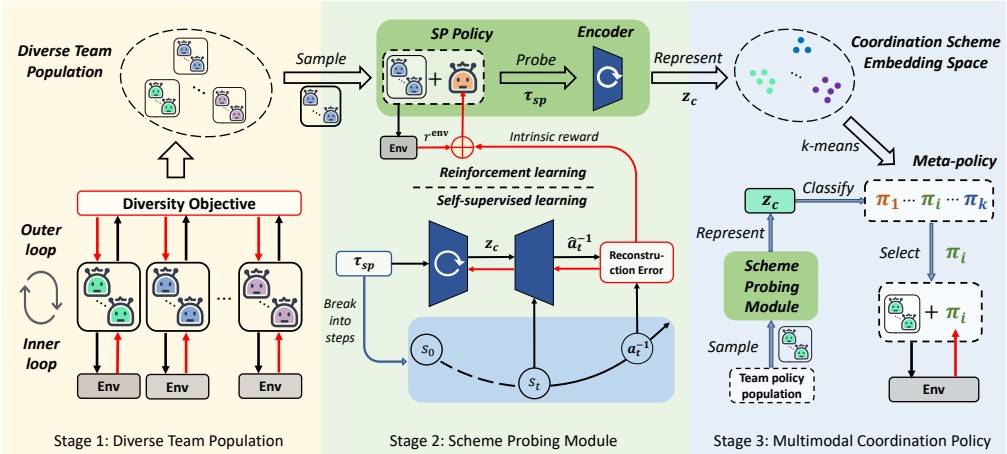

Figure 2: Overall framework of CSP, where SP is short for "scheme probing".

---

**Algorithm 1** CSP: Brief Training Process

---

1: Initialize training population $\Pi_{\text{train}} = \{\boldsymbol{\pi}^1, \ldots, \boldsymbol{\pi}^M\}$, scheme probing module $\langle E_c, D_c\rangle, \pi_{\text{sp}}$ and multimodal meta-policy $\boldsymbol{\pi}_{\text{meta}} = \{\boldsymbol{\pi}_1, ..., \boldsymbol{\pi}_k\}$.
2: Train each team policy $\boldsymbol{\pi}^i \in \Pi_{\text{train}}$ for quality and diversity.          ▷ (Fig. 2, Stage 1)
3: Train $\langle E_c, D_c\rangle$ to represent teammates, and $\pi_{\text{sp}}$ to probe teammates.          ▷ (Fig. 2, Stage 2)
4: Represent teams in $\Pi_{\text{train}}$ and discover coordination schemes as $k$ clusters.    ▷ (Fig. 2, Stage 3)
5: Train $\pi_{\text{meta}}$ for generalization by each sub-policy acquiring a unique scheme.
6: **return** $\pi_{\text{meta}}$

---

## 4.1 DIVERSE TEAM POPULATION

We first need to use a set of diverse team policies to simulate the coordination schemes required for the training and evaluation phases. Instead of letting each team acquire diversity with different initializations (Carroll et al., 2019a; Strouse et al., 2021) alone, we propose the Soft-Value Diversity (SVD) objective and an alternate optimization process to gain diversity explicitly. We maintain a population with $M$ independent multi-agent teams, each learning with an MARL method (VDN (Sunehag et al., 2018) in this work). While each team performs independent "inner loop" learning, the population periodically performs centralized "outer loop" updates for all teams to maximize SVD:

$$J_{\text{SVD}}(\{\theta_i\}_{i=1}^M) = \mathbb{E}_{\boldsymbol{\tau},\boldsymbol{a}} \left[ \sum_{i=1}^{M} \|\overline{D}(\boldsymbol{\tau},\boldsymbol{a}) - D_i(\boldsymbol{\tau},\boldsymbol{a})\|_2^2 \right], \qquad (1)$$

where $\theta_i$ denotes the Q-network's parameters for the $i$-th team's policy, $\boldsymbol{\tau}$ and $\boldsymbol{a}$ denote individual observation and action, $D_i(\boldsymbol{\tau},\boldsymbol{a}) = \frac{\exp\left(Q_{\theta_i}(\boldsymbol{\tau},\boldsymbol{a})\right)}{\sum_{\boldsymbol{a}' \in \mathcal{A}} \exp\left(Q_{\theta_i}(\boldsymbol{\tau},\boldsymbol{a}')\right)}$ is the normalized value estimation obtained by performing the Boltzmann softmax operator (Asadi & Littman, 2017), and $\overline{D}(\boldsymbol{\tau},\boldsymbol{a}) = \frac{1}{M}\sum_{i=1}^M D_i(\boldsymbol{\tau},\boldsymbol{a})$ is the average observation-action value estimation for all policies in the population $\Pi$. We use the Monte Carlo method to estimate the expectation based on recent collected trajectories in the replay buffer. In this stage, all agents (i.e., $\mathcal{N} = G^1 \cup G^{-1}$) for each team are trained jointly, but we only store $\pi^{-1}$ corresponding to $G^{-1}$ for further use.

Maximizing SVD encourages teams within the population to have different value estimations of each observation-action pair by increasing the gap between their current values and the mean. This diversity in value estimation reflects various local optima caused by different joint decisions in multi-agent scenarios, where the same action will have different outcomes when teammates' actions change. The "inner" and "outer" loops in the framework are designed to improve individual team performance and diversity among different teams. We set a hyper-parameter to control the learning frequency of these two loops, which is a trade-off between the quality and diversity of policies in the population. We obtain $\Pi_{\text{train}}$ and $\Pi_{\text{eval}}$ independently via this process.

## 4.2 SCHEME PROBING MODULE

After having a diverse team population $\Pi_{\text{train}}$, we train the scheme probing module upon it to efficiently reveal and represent different teams' coordinating policies. It has two main parts. One is the scheme probing policy $\boldsymbol{\pi}_{\text{sp}}$ which interacts with a team for an entire episode and gathers the trajectory $\tau_{\text{sp}} = \{s_t, \boldsymbol{a}_t\}_{t=0}^{H}$ that reveals the current teammates' coordination scheme. The other is the team-dynamics autoencoder $\langle E_c, D_c \rangle$, which learns a representation $z_c$ of team policies based on $\tau_{\text{sp}}$ in a self-supervised manner. Start from here, we only train policies for $G^1$ and let $G^{-1}$ switch within $\Pi_{\text{train}}$. The teammates can thus be viewed as a non-stationary part of the environment.

**Team-dynamics Autoencoder.** The encoder $E_c$ is parameterized as an LSTM (Hochreiter & Schmidhuber, 1997) which takes as input $\tau_{\text{sp}}$ and outputs an embedding vector $z_c$. The decoder $D_c$ is a feed-forward network that takes as input both $z_c$ and current state $s_t$, and predicts teammates' next joint action distribution $\boldsymbol{a}_t^{-1}$. Formally,

$$z_c = E_c\left(\tau_{\text{sp}};\ \theta_c\right), \quad \hat{\boldsymbol{a}}_t^{-1} = D_c\left(\cdot \mid s_t, z_c;\ \phi_c\right). \tag{2}$$

The parameters $\theta_c$ and $\phi_c$ are jointly optimized to minimize the cross entropy loss (i.e., reconstruction error in Fig. 2) of $\hat{\boldsymbol{a}}_t^{-1}$ and $\boldsymbol{a}_t^{-1}$ averaged over the entire trajectory $\tau_{\text{sp}}$:

$$\mathcal{L}_{\text{tot}} = \frac{1}{H}\sum_{t=1}^{H}\mathcal{L}_{\text{pred}}(t) = -\frac{1}{H}\sum_{t=1}^{H}\log D_c(\boldsymbol{a}_t^{-1} \mid s_t, z_c;\ \phi_c). \tag{3}$$

We call this approach *team-dynamics reconstruction* for the following reasons. Firstly, $D_c$ receives no historical information, so it cannot infer a team's behaviors based on the temporal ordering of states and joint actions. In this case, $E_c$ is forced to embed information about this team's dynamics into $z_c$ to make a good reconstruction. Additionally, since $s_t$ is already input of $D_c$, $E_c$ has no motive to embed any information about states. Thus, the embedding $z_c$ contains only information about the policy and not the environment, making it a compressed and precise representation of a team.

**Scheme Probing Policy.** Instead of collecting information passively, we expect $\boldsymbol{\pi}_{\text{sp}}$ to actively guide teammates' behaviors and reveal their coordination schemes. To this end, we introduce the reconstruction loss above at each timestep as an intrinsic reward for $\boldsymbol{\pi}_{\text{sp}}$, which is added to the original environmental reward $r_t^{\text{env}}$:

$$r_t' = r_t^{\text{env}} + \alpha\mathcal{L}_{\text{pred}}(t), \tag{4}$$

where $\alpha$ is an adjustable hyperparameter to achieve a trade-off between coordination performance and information gain through probing. This additional term encourages $\boldsymbol{\pi}_{\text{sp}}$ to explore states with the large behavioral uncertainty across different kinds of teammates, which is considered helpful in determining their identities and representing their coordination policies.

## 4.3 MULTIMODAL COORDINATION POLICY

When the scheme probing module is well trained, it is fixed and used to guide downstream scheme-specific control. Common context-based methods (Hausman et al., 2018; Yang et al., 2020) use embeddings as augmentation of the agent's input space. However, in scenarios of coordination generalization, the behaviors under different coordination schemes can vary greatly and conflict with each other. Such processes require a single network to acquire multimodal behaviors as the context changes, which increases learning complexity and instability. As a comparison, we use embeddings to automatically group similar team policies to discover different schemes and solve each distinct group with an independent sub-policy to avoid conflicts.

**Scheme Discovery.** We first use the learned scheme probing module to probe and represent all teams in $\Pi_{\text{train}}$ $N$ times, which generates an embedding set $Z$ of size $|Z| = NM$. Repeating it $N$ times is to get the distribution of a team's representation rather than a single sample point when the environment and policy are stochastic. Then we perform $k$-means clustering based on Euclidean distances on $Z$ to get $k$ clusters with centers $\boldsymbol{\mu} = \langle \mu_1, \ldots, \mu_k \rangle$. We use the Silhouette method (Rousseeuw, 1987) to automatically determine the most suitable $k$, and details can be found in App. A.2.

These clusters reflect the natural structure of coordination schemes, as behaviors under the same scheme should not vary too much to ensure coordination. The number of clusters $k$ will be approximately equal to the number of environmental coordination schemes $|C|$ if $\Pi_{\text{train}}$ already covers all possible schemes. So enlarging the size of $\Pi_{\text{train}}$ will not infinitely increase the learning complexity.

**Meta-Policy Learning.** We initialize a multimodal meta-policy with $k$ sub-policies, where each sub-policy takes local observation history as input:

$$\boldsymbol{\pi}_{\mathrm{meta}} = \{\boldsymbol{\pi}_i(a \mid \tau) \mid i = 1, \ldots, k\}. \tag{5}$$

During training or deploying, when confronted with a team, we first let the scheme probing policy $\boldsymbol{\pi}_{\mathrm{sp}}$ interact with it to get trajectory $\tau_{\mathrm{sp}}$ and get the representation $z_c = E_c(\tau_{\mathrm{sp}})$. Then, we classify the team into one of the $k$ classes (discovered schemes) according to the distance of $z_c$ and all cluster centers $\boldsymbol{\mu}$. Once we have done the classification, we choose the corresponding sub-policy to do the rest of the control. Formally,

$$\boldsymbol{\pi}_{\mathrm{meta}}(a \mid \tau, \tau_{\mathrm{sp}}) = \boldsymbol{\pi}_{i^*}(a \mid \tau), \quad \text{where } i^* = \arg\min_i \|E_c(\tau_{\mathrm{sp}}) - \mu_i\|_2^2. \tag{6}$$

This structural design takes advantage of multimodality to be highly expressive, allowing end-to-end learning of several vastly distinct coordination schemes simultaneously without affecting each other. Moreover, each sub-policy $\boldsymbol{\pi}_i(a|\tau)$ only needs to acquire a unique and stationary coordination scheme. Compared to common context-based methods that use a single policy $\boldsymbol{\pi}(a \mid \tau, z)$ to acquire all schemes, it reduces learning complexity and improves stability.

## 5 EXPERIMENTS

In the section, we design experiments to answer the following questions: (1) How well does CSP perform when generalizing to unknown partners in multiple complex scenarios (Sec. 5.1)? (2) Can the proposed scheme probing process get meaningful and distinguishable representations (Sec. 5.2)? (3) Does the team population really have multiple coordination schemes (Sec. 5.3)? (4) What is the impact of each component of CSP (Sec. 5.4)?

We select four multi-agent cooperative environments with six scenarios as benchmarks: Level-based Foraging (LBF) (Papoudakis et al., 2021b) needs agents to find food randomly distributed on the map and eat it together. Predator-Prey (PP) (Böhmer et al., 2020) is a more challenging version of LBF that allows each prey to move randomly at each timestep. Overcooked (Strouse et al., 2021) requires two players coordinate to cook and deliver food to target locations. We use a standard layout *Coordination Ring* and a modified layout *Forced Coordination Hard* that enhances penalties for miscoordination. SMAC (Samvelyan et al., 2019) is a commonly used MARL benchmark that focuses on coordinated micromanagement. We use a standard map *1c3s5z* and a customized map *Fork* specially designed with multiple coordination schemes.

Six state-of-the-art approaches from three related fields are chosen for comparison. (1) Meta-learning: PEARL (Rakelly et al., 2019) uses recently collected context to infer a probabilistic variable describing the task. (2) Policy representation: LIAM (Papoudakis et al., 2021a) is an MARL method that predicts teammates' current behaviors based on local observation history, and FIAM (Papoudakis et al., 2021a) enhances LIAM by replacing local observations with global states. (3) Ad hoc teamwork: PBT (Carroll et al., 2019b) uses domain randomization to train an ad hoc agent with a population of pre-trained partners. FCP (Strouse et al., 2021) draws on the practice of FSP (Heinrich et al., 2015) in games by constructing training sets with teammates' historical checkpoints. ODITS (Gu et al., 2022) applies a centralized "teamwork situation encoder" for end-to-end learning.

Notice that PEARL, PBT, FCP, and ODITS are originally designed for single-agent settings. To make them compatible with multi-agent scenarios, we implement them upon an MARL framework VDN (Sunehag et al., 2018) that follows the CTDE paradigm to learn team policies for $G^1$. Other multi-agent methods (i.e., CSP, LIAM, and FIAM) also choose VDN as their base MARL framework for a fair comparison. In addition, the original versions of LIAM and FIAM did not use populations and only did self-play during training, while PBT, FCP and ODITS used several teams with different random seeds and initializations to construct training populations, which may be unstable and difficult to compare. To standardize comparison, we reuse the same $\Pi_{\mathrm{train}}$ generated in CSP's stage 1 for all baselines, thus reducing the chance of outcome due to randomness of population generation. More details of each scenario and baseline are shown in App. B-C.

### 5.1 EVALUATION OF PERFORMANCE

We first investigate how CSP and other baselines perform when coordinated with unknown diverse teammates. Fig. 3 shows the average performance of coordinating with all teams in $\Pi_{\mathrm{eval}}$ during

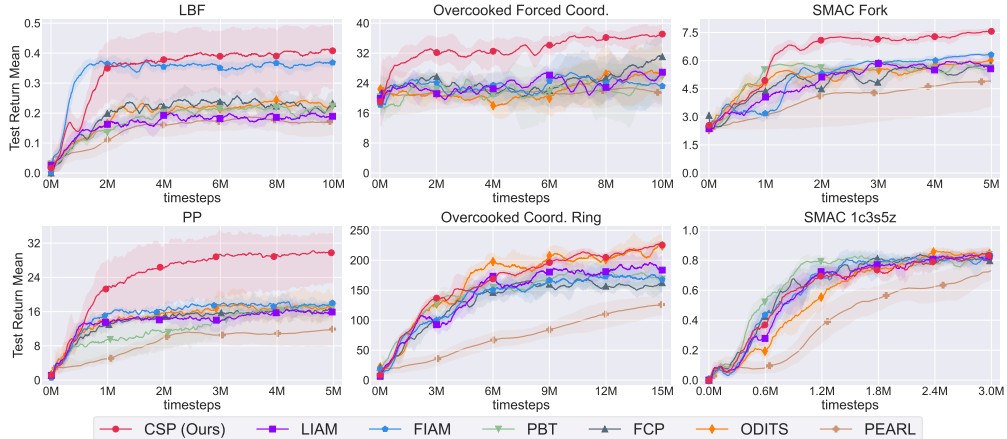

Figure 3: Mean generalization performance on $\Pi_{\text{eval}}$ of all methods.

training. Since CSP always collects a probing trajectory $\tau_{\text{sp}}$ before adapted coordination, we use the average score of the two trajectories as CSP's performance metric for fair comparisons. All experiments are done with five random seeds, and all curves are marked with 95% confidence intervals in the shaded area.

In general, CSP outperforms PEARL, PBT, FCP, and LIAM consistently and is better or at least comparable to FIAM and ODITS according to the tasks. In LBF, where each agent has a strictly limited observation range, FIAM has a clear advantage because it utilizes global information at each timestep to help make decisions. The comparable result of CSP shows that it is possible to achieve near-optimal control with pre-identified coordination scheme information alone, eliminating the need for global states throughout the entire episode. In PP, Overcooked, and SMAC Fork, the cooperative tasks are challenging and contain several vastly different coordination schemes, which require the coordinator to change its behaviors aggressively and precisely. All baselines here have a clear gap with CSP. We believe the superiority comes from the fact that CSP uses multimodality to isolate the expression of different coordination schemes to avoid mutual interference. As a comparison, in SMAC 1c3s5z, CSP and all baselines converge to roughly equivalent performance. The reason is that without a specially designed symmetry like Fork, the optimal policy in this task is reflected in focusing fire on the enemy and retreating when its health is low. The optimal decision under each state is sure with no ambiguity of different coordination schemes, and all methods acquire this optimal policy after training long enough. Although PEARL uses additional context data, it performs worse than CSP. We believe it is because PEARL learns to encode context into a meaningful variable in an end-to-end manner to directly maximize cumulative reward, which is relatively inefficient. It is worth noting that CSP trains $k$ sub-policies but still has comparable or even better sample efficiency to baselines, verifying that isolating different schemes stabilizes training and avoids conflict schemes from affecting each other.

## 5.2 SCHEME REPRESENTATION ON SMAC

A meaningful and distinguishable representation of coordination schemes is the basis of CSP's adaptation ability. To demonstrate this, we visualize the embeddings of our scheme probing module with experiments on SMAC Fork (Fig. 4). We let our scheme probing module interact with all teams in $\Pi = \Pi_{\text{train}} \cup \Pi_{\text{eval}}$ for $64$ episodes. One of the baselines, LIAM, which does agent modeling at every timestep, also runs in parallel as a comparison.

Let's first make a brief understanding of the Fork task. There are two symmetrical points at *Up* and *Down* sides guarded by several enemies. If all the teammates choose to attack the same point, they will have enough force to eliminate all the enemies there and get a high reward. Otherwise, neither point will have a firepower advantage, and all the teammates will be destroyed and fail. Therefore, we can roughly claim that there are two basic coordination schemes $C = \{Up, Down\}$ in this task. Screenshots at timesteps $10$ and $30$ (Fig. 4a-b) indicate the early and middle stages of one episode. Asynchronous decision makings occur in (a) and (b) for $G^1$ and $G^{-1}$ respectively to select a direction (*Up* or *Down*) to attack, and coordination succeeds if their choices are the same. Since $G^{-1}$ may be

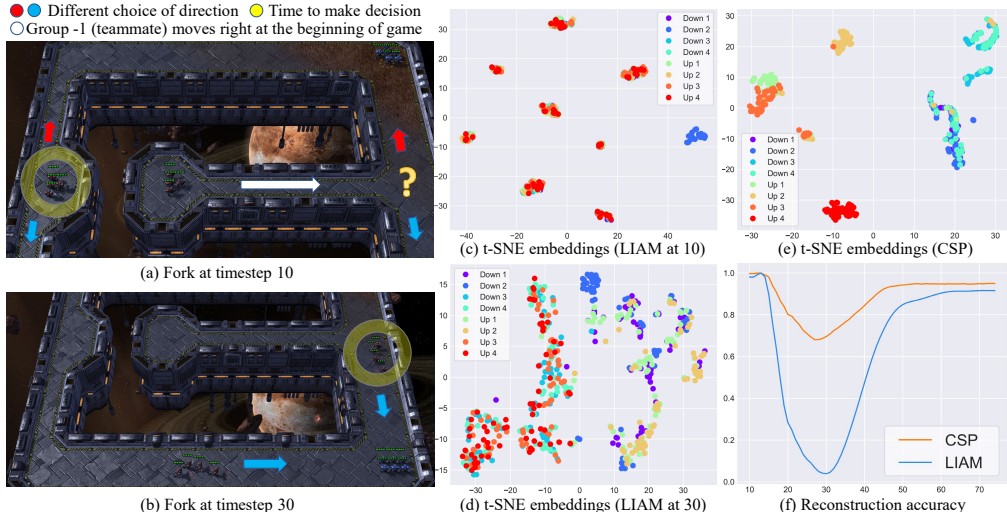

Figure 4: (a)(b) SMAC Fork at timesteps 10 and 30, when $G^1$ (left) and $G^{-1}$ (right) make decisions respectively to attack enemies at point *Up* (red arrow) or *Down* (blue arrow). (c)(d) LIAM's embeddings of different teams at timesteps 10 and 30. (e) CSP's embeddings of different teams. (f) CSP and LIAM's mean reconstruction accuracy of $G^{-1}$'s joint action $\boldsymbol{a}_t^{-1}$ throughout one episode.

controlled by various policies in $\Pi$, which will make different decisions, $G^1$ has to make the right choice based on the partners it meets.

The t-SNE projection of CSP's scheme embeddings is shown in Fig. 4e, where different colors represent different teams in $\Pi$ (named by their scheme observed) and each point represents a single run. There are two main phenomenons: (1) Each color forms a relatively compact cluster. (2) Clusters with similar colors (lighter or darker) tend to be close together, while deeper and lighter colors are farther apart. The former indicates that CSP's representations are highly consistent with low variance, which is beneficial for stabilizing downstream MARL learning. The latter shows that the embedding space holds semantically meaningful information, where teams with similar coordination schemes can be packed together. As a comparison, LIAM's embeddings at timesteps 10 and 30 are shown in Fig. 4c-d. We can observe that different teams are mixed up at the early stage, and a few local clusters emerge as time goes by but still cannot distinguish well between teams. As illustrated in the screenshots (white arrow), $G^{-1}$ will always move right in the first few dozen steps, so the trajectories of different teams during this period are similar and hold insufficient information. It is impossible for LIAM to distinguish teams based on it, let alone make the right choice in the beginning.

Fig. 4f presents the mean reconstruction accuracy of $G^{-1}$'s joint actions throughout one episode. We can observe that CSP has consistently higher accuracy than LIAM, especially at timesteps close to 30. As described above, the observation segment before is not informative enough, in which case LIAM will fail to predict teammate behaviors when sudden uncertainty occurs. By contrast, CSP has a comprehensive view of the teammates it coordinates with after the probing phase in advance, so it can fully guarantee its scheme prediction throughout the coordination phase. More results on other benchmarks are shown in App. D.1.

### 5.3 SCHEME DIVERSITY

To verify populations generated in stage 1 do hold multiple coordination schemes, we perform Cross-Play (Hu et al., 2020) experiments on Overcooked's Coordination Ring and SMAC Fork (Fig. 5a). Teams in $\Pi = \Pi_{\text{train}} \cup \Pi_{\text{eval}}$ are paired to play the role of $G^1$ and $G^{-1}$ for all combinations.

Firstly, we can find that values on the diagonal from the top left to the bottom right corner are generally larger than others. This indicates that each team coordinates well with itself, as they are always in the same coordination scheme. As a comparison, the relatively lower performance of other points indicates the inability to coordinate across different schemes. Interestingly, the performance

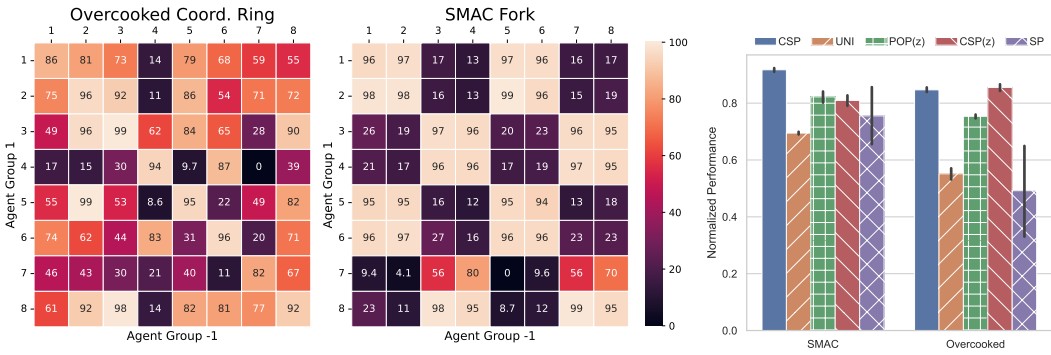

(a) Cross-Play performance of full population $\Pi$, normalized to $[0, 100]$.     (b) Ablations.

Figure 5: (a) The scores of cross-play on the Overcooked Coordination Ring and SMAC Fork benchmarks. (b) Final generalization performance of CSP and ablations for each component.

on SMAC Fork shows that it has two basic coordination schemes, where each team can coordinate with exactly half of the teams in $\Pi$ that have the same coordination scheme. This phenomenon is aligned with our understanding of the task, as described above in Sec. 5.2. More results on other benchmarks are provided in App. D.2.

## 5.4 ABLATION STUDY

We perform ablations on SMAC and Overcooked to demonstrate the importance of CSP's different components: UNI removes multimodality and only has a single sub-policy $\pi(\tau)$; POP(z) uses a single context-based policy $\pi(\tau, z)$ instead of sub-policies; CSP(z) extends each sub-policy to be $\pi_i(\tau, z)$; SP does not use $\Pi_{\text{train}}$ and learns with self-play alone. We report the mean generalization performance with $\Pi_{\text{eval}}$ and error bars indicating the $95\%$ confidence interval as shown in Fig. 5b.

Firstly, we can observe that SP generally does the worst and has a large variance. This means that without being exposed to diverse partners during training, the policy can only find a single coordination scheme and is hard to generalize. UNI has a lower variance but still performs poorly, which indicates that using domain randomization helps make the policy robust to partner changes, but it cannot be specialized to each scheme without a specially designed adaptability module. The relatively lower performance of POP(z) compared to CSP confirms our claim earlier that it is difficult to make adaptations within a single network based on different scheme embedding. In complex multi-agent scenarios, each coordination scheme can imply highly different and even opposite behaviors, where using multiple sub-policies helps to isolate these schemes and makes each consistent. Finally, CSP(z) does not outperform the original version. This phenomenon shows that our scheme grouping process already fully uses the information contained in the embedding. Further use of it as additional input no longer results in a boost but may increase learning difficulty.

## 6 CLOSING REMARKS

To achieve generalizable coordination in complex multi-agent scenarios and address limitations of ad hoc teamwork, this paper considers learning a coordination scheme probing module for teammates recognition and a meta-policy consisting of multiple sub-policies for few-shot coordination with unseen diverse teams. With the help of this probing module, we can reduce few-shot coordination to a multi-task RL problem by clustering the representation space. A multimodal policy is then end-to-end trained to solve it directly. Sufficient experiments compared against strong baselines on various benchmarks validate the effectiveness of our proposed method. We point out two limitations and interesting future work: (1) Co-evolution of the population would be a more general interaction setting, such as Quality Diversity (Parker-Holder et al., 2020). (2) Effective knowledge transfer between submodules, such as Soft Modularization (Yang et al., 2020), could be considered to improve sample efficiency. Finally, instead of training alongside artificial agents, we also hope to study the human-in-the-loop setting to adapt to people's dynamic needs and preferences.

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

# A    IMPLEMENTATION DETAILS OF CSP

## A.1    ALGORITHMS

We now give more a detailed pseudocode corresponding to different stages of CSP.

---

**Algorithm 2** CSP: Training Stage 1

---

1: Initialize teammate policies $\Pi = \{\boldsymbol{\pi}^1, \ldots, \boldsymbol{\pi}^M\}$
2: **while** not done **do**
3:     **for** inner loops **do**
4:         **for** $\boldsymbol{\pi}^j \in \Pi$ **parallel do**
5:             Control $G^1 \cup G^{-1}$ with $\boldsymbol{\pi}^j$. Train $\boldsymbol{\pi}^j$ with any MARL algorithm
6:         **end for**
7:     **end for**
8:     **for** outer loops **do**
9:         Update all $\boldsymbol{\pi}^i \in \Pi$ to maximize $J_{\text{SVD}}$    (Eq. 1)
10:     **end for**
11: **end while**
12: **return** $\Pi$

---

**Algorithm 3** CSP: Training Stage 2

---

1: **Input:** Teammate policy population $\Pi_{\text{train}}$
2: Initialize scheme probing policy $\boldsymbol{\pi}_{\text{sp}}$, autoencoder $\langle E_c, D_c \rangle$, and replay buffer $B$
3: **while** not done **do**
4:     **for** $\boldsymbol{\pi}^j \in \Pi$ **do**
5:         Control $G^1$ with $\boldsymbol{\pi}_{\text{sp}}$ and $G^{-1}$ with $\boldsymbol{\pi}^j$ to generate $\tau_{\text{sp}} = \{o_t, a_t\}_{t=0}^H$
6:         $B \leftarrow B \cup \{\tau_{\text{sp}}\}$
7:         Sample trajectories $\mathcal{D}$ from $B$
8:         Compute $\{\mathcal{L}_{\text{pred}}(t)\}_{t=0}^H$ based on $\mathcal{D}$ and update $\langle E_c, D_c \rangle$    (Eq. 3)
9:         Train $\boldsymbol{\pi}_{\text{sp}}$ by any MARL algorithm, with $r'_t = r_t^{\text{env}} + \alpha \mathcal{L}_{\text{pred}}(t)$    (Eq. 4)
10:     **end for**
11: **end while**
12: **return** $\pi_{\text{sp}}, E_c$

---

**Algorithm 4** CSP: Training Stage 3

---

1: **Input:** $\Pi_{\text{train}}, \pi_{\text{sp}}, E_c$
2: Initialize team embedding set $Z$; multimodal meta-policy $\pi_{\text{meta}} = \{\pi_1, \ldots, \pi_k\}$
3: **for** $\pi^j \in \Pi_{\text{train}}$ **do**
4:     **for** $n$ in $N$ **do**
5:         Control $G^1$ with $\boldsymbol{\pi}_{\text{sp}}$ and $G^{-1}$ with $\boldsymbol{\pi}^j$ to generate $\tau_{\text{sp}} = \{o_t, a_t\}_{t=0}^H$
6:         Get team embedding $z_c = E_c(\tau_{\text{sp}}), Z \leftarrow Z \cup \{z_c\}$
7:     **end for**
8: **end for**
9: Compute best $k$ with the Silhouette method on $Z$    (Eq. 8)
10: Compute centers $\boldsymbol{\mu} = \langle \mu_1, \ldots, \mu_k \rangle$ with k-means clustering on $Z$
11: **while** not done **do**
12:     **for** $\pi^j \in \Pi_{\text{train}}$ **do**
13:         Control $G^1$ with $\boldsymbol{\pi}_{\text{sp}}$ and $G^{-1}$ with $\boldsymbol{\pi}^j$ to generate $\tau_{\text{sp}} = \{o_t, a_t\}_{t=0}^H$
14:         Get team embedding $z_c = E_c(\tau_{\text{sp}})$
15:         Pick sub-policy $\pi_{i^*}$ based on $i^* = \arg\min_i \|z_c - \mu_i\|_2^2$    (Eq. 6)
16:         Control $G^1$ with $\boldsymbol{\pi}_{i^*}$ and $G^{-1}$ with $\boldsymbol{\pi}^j$. Train $\boldsymbol{\pi}_{i^*}$ with any MARL algorithm
17:     **end for**
18: **end while**
19: **return** $\boldsymbol{\mu} = \langle \mu_1, \ldots, \mu_k \rangle, \pi_{\text{meta}} = \{\pi_1, \ldots, \pi_k\}$

---

---

**Algorithm 5** CSP: Deploying

1: **Input:** $\pi_{\text{sp}}, E_c, \boldsymbol{\mu} = \langle \mu_1, \ldots, \mu_k \rangle, \pi_{\text{meta}} = \{\pi_1, \ldots, \pi_k\}$, new policy $\pi^{\text{new}}$ to adapt to
2: Control $G^1$ with $\boldsymbol{\pi}_{\text{sp}}$ and $G^{-1}$ with $\boldsymbol{\pi}^{\text{new}}$ to generate $\tau_{\text{sp}} = \{o_t, a_t\}_{t=0}^H$
3: Get team embedding $z_c = E_c(\tau_{\text{sp}})$
4: Pick sub-policy $\pi_{i^*}$ based on $i^* = \arg\min_i \|z_c - \mu_i\|_2^2$   (Eq. 6)
5: Execute $\pi_{i^*}$ to coordinate with $\pi^{\text{new}}$

---

## A.2 SILHOUETTE METHOD

We use the Silhouette method (Rousseeuw, 1987) to automatically determine the most suitable $k$ for the embedding set $Z$. The intuition is that a higher Silhouette value for a data point indicates that this point is placed in the correct cluster. Therefore, a cluster number $k$ with the highest mean Silhouette value for all points in a dataset is desirable. Concretely, for the embedding set $Z$, the Sihouette value $SV(i)$ for each data point $i$ that belongs to $I$-th cluster $Z_I$ is defined as:

$$SV(i) = \begin{cases} \dfrac{d_{\text{out}}(i) - d_{\text{in}}(i)}{\max\{d_{\text{out}}(i), d_{\text{in}}(i)\}} & , \quad \text{if } |Z_I| > 1, \\ 0 & , \quad \text{if } |Z_I| = 1, \end{cases} \tag{7}$$

where $d_{\text{in}}(i)$ denotes the mean distance between data point $i$ and other points in the same cluster, and $d_{\text{out}}(i)$ denotes the smallest mean distance of data point $i$ to all points in any other cluster. They are defined as $d_{\text{in}}(i) = \frac{1}{|Z_I|-1} \sum_{j \in Z_I, i \neq j} d(i,j)$ and $d_{\text{out}}(i) = \min_{I \neq J} \frac{1}{|Z_J|} \sum_{j \in Z_J} d(i,j)$, where $d(i,j)$ is the Euclidean distance between data points $i$ and $j$. We implemented an automatic $k$ selection approach by linear searching for the maximum mean Silhouette value of all data points:

$$k^* = \arg\max_{k \leq M} \frac{1}{|Z|} \sum_{i \in Z} SV(i). \tag{8}$$

## A.3 HYPERPARAMETERS AND ARCHITECTURE

In Stage 1, "inner loops" and "outer loops" are set to 32 and 5, respectively. MARL refers to any multi-agent reinforcement learning method, and we choose VDN (Sunehag et al., 2018) here. Population size $M = |\Pi_{\text{train}}| = |\Pi_{\text{eval}}|$ is 4 for all scenarios but 1c3s5z, which is set to 5 instead. In Stage 2, $\alpha$ is set to $1 \times 10^{-6}$ and $|\mathcal{D}|$ is set to 32. In Stage 3, $N$ is set to 64.

Details of neural network architectures used by CSP are provided in Fig. 6. Each policy used in CSP (i.e., teammate policy $\pi^i \in \Pi$, probing policy $\pi_{\text{sp}}$, and each sub-policy in $\{\pi_1, \ldots, \pi_k\}$) is a form of "GRU policy" with local observation history as input and outputs value estimation across its action space. The MARL framework VDN adds up all agents' local utility $q_i$ to form $Q_{\text{tot}}$, which is updated to approximate the global discounted return.

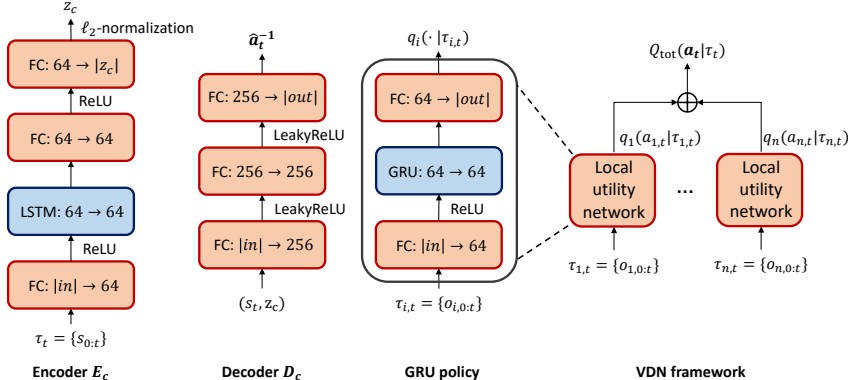

Figure 6: Network architectures of CSP, where $o_{i,t}$ and $a_{i,t}$ represent the local observation and action for agent $i$ at timestep $t$, and $\tau_{i,t}$ is the trajectory of agent $i$'s local observations until timestep $t$.

## B    DETAILS OF ENVIRONMENTS

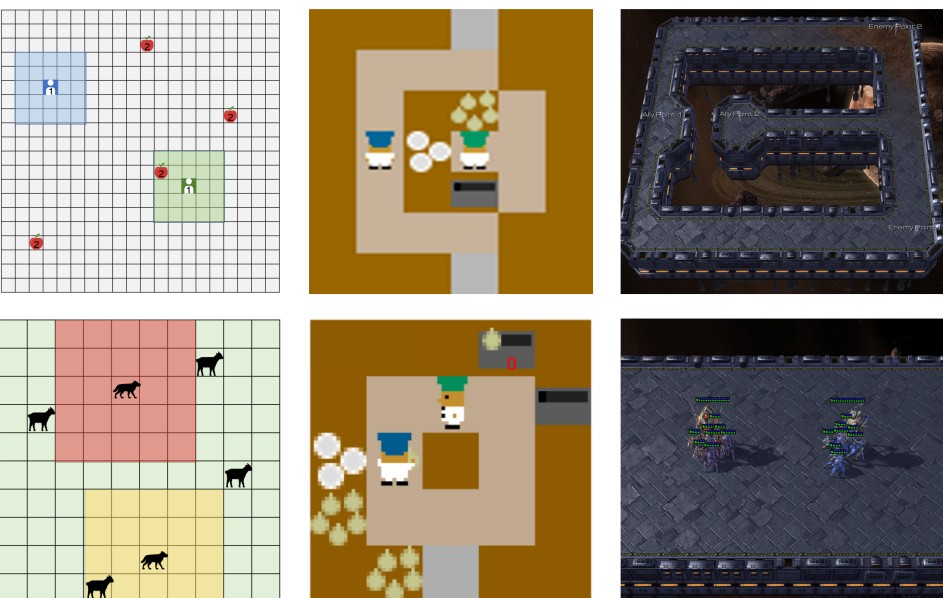

Figure 7: All the environments used in this paper. The first row from left to right: Level-based Foraging (LBF), Overcooked Forced Coordination Hard, SMAC Fork. The second row from left to right: Predator-Prey (PP), Overcooked Coordination Ring, SMAC 1c3s5z.

**Level-based Foraging (LBF) (Papoudakis et al., 2021b).** We use an instance of LBF, where the environment is a $20 \times 20$ grid world with 2 agents and 4 food. Each agent has a self-centered $5 \times 5$ observation range and a discrete action space for moving in four directions and collecting food. The goal of each agent is to collect all the food on the map. When food is collected, the environment returns a shared reward proportional to the food level with a total reward normalized to 1. An episode terminates if all the food is collected or reaches 50 total timesteps. To enhance the requirement for cooperation, we set the extra constraint that each food can only be collected if both agents are adjacent and perform the "collect" action simultaneously. We let CSP or baselines control one player and teammates from $\Pi$ control another. The variability of coordination schemes in this environment is reflected in the order of eating all 4 food.

**Predator-Prey (PP) (Böhmer et al., 2020).** This environment can be considered a more complex version of LBF, where 2 predators with a $5 \times 5$ observation range are expected to hunt 4 prey in a $10 \times 10$ grid world. An episode ends when all prey is captured, or 200 timesteps have passed. The extra difficulty comes from the fact that each prey moves randomly throughout the game, so predators must constantly jointly chase the prey. We simplify the task by removing the "capture" action, and predators can capture prey with only a siege. Therefore, there will be no miss-capturing punishment. The coordination schemes in this environment are reflected in the order of chasing prey and the respective division of labor in the roundup of prey.

**Overcooked (Strouse et al., 2021).** There are 2 agents in the environment sharing a 6-dimensional discrete action space: moving in four directions, interacting with the object facing, and doing nothing. Cooking a dish requires a series of actions and a waiting period. Delivering a dish requires picking it up, moving to the correct delivery point, and putting it down. The goal of both agents is to complete as many delivery orders as possible within 400 timesteps. We set CSP or baselines to control the blue player and teammates from $\Pi$ to control the green player. In the layout Coordination Ring (lower), the passage is narrow, and the two agents may clash in their pathfinding. In the layout Forced Coordination Hard (upper), only the green agent can touch the cookware, and only the blue agent can reach the delivery point. Coordination is forced in this layout since no agent can finish the task alone, and they have to adapt to teammates' preferences.

**SMAC (Samvelyan et al., 2019).** It is widely used as a multi-agent benchmark for its high complexity of control. Each agent can move in four cardinal directions, stop, do nothing, or select an entity to interact (heal or attack according to its type) at each timestep. Therefore, if there are $n_a$ allies and $n_e$ enemies in the map, the action space for each unit contains $n_a + n_e + 6$ discrete actions, and the joint action space size is $(n_a + n_e + 6)^{n_a}$. The map 1c3s5z is a standard map that requires control of 9 agents. We set CSP or baselines to control the first 4 agents, and teammates in $\Pi$ control the reset 5 agents.

We specially designed a map called *Fork* (Fig. 8) for this work which requires strong coordination and has very different coordination modes. It has 2 ally spawn points on the left and middle left sides of the map and 2 enemy spawn points on the upper right and lower right corners. At the beginning of an episode, each ally spawn point generates 4 marines (long-range attack unit), and each enemy spawn point generates 6. We let CSP or baselines control allies at point 1 and teammates from $\Pi$ to control allies at point 2. If both groups choose to attack the same group of enemies, it will be 8 versus 6 and is easy to win. By contrast, if two groups of teammates attack different groups of enemies, respectively, it will be 4 versus 6, and neither group will be able to defeat. Thus, intuitively there are two main kinds of cooperation modes, which we refer to as *Up* and *Down* in this work, indicating attack enemies in the corresponding direction first. The population $\Pi$ is trained with CSP's stage 1. In order to make the population unbiased in attacking directions, we manually pick 8 policies from the original population with 4 *Up*s and 4 *Down*s to form the balanced population.

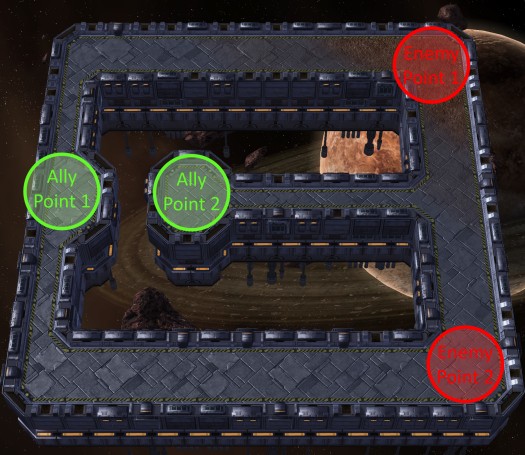

Figure 8: New SMAC map Fork.

## C    DETAILS OF BASELINES

We compare CSP against six baselines. All of them are implemented with a similar GRU agent and VDN framework as CSP (Fig. 6), except that the input of GRU agents is $\tau_t = \{(o, z_c)_{0:t}\}$ which has additional embedding $z_c$ at each timestep, so we mainly focus on how they build $z_c$ with components shown in Fig. 9.

**PEARL (Rakelly et al., 2019).** This baseline comes from single-agent and meta-learning settings. It aims to represent the environments by hidden representations. It utilizes the history-data as context to inference the feature of the environment, which is modeled by a product of Gaussians. Since we use the Dec-POMDP setting, the PEARL module is adopted and optimized on each individual policy.

**Population based trainin (PBT).** This baseline uses simple domain randomization to train $G^1$ against all the teams in $\Pi_{\text{train}}$. The training set $\Pi_{\text{train}}$ is required from CSP's stage 1. We think fixing this population increases the fairness of the comparison.

**Ficticious co-play (FCP) (Strouse et al., 2021).** This baseline uses similar domain randomization approach like PBT to train $G^1$, except that the training population is an extended version of $\Pi_{\text{train}}$. Checkpoints at one-third and two-thirds of the total training timesteps are added to $\Pi_{\text{train}}$, indicating teammates with different levels of ability.

**Local information agent model (LIAM) (Papoudakis et al., 2021a).** This baseline equips each agent with an encoder-decoder structure to predict other agents' observations $o_t^{-1}$ and actions $a_t^{-1}$ at current timestep based on its own local observation history $\tau_t = \{o_{0:t}\}$. The encoder and decoder are optimized to minimize the mean square error of observations plus the cross-entropy error of actions. The original version of LIAM considers only a single controllable agent, and predictions are made upon this agent's local observation. To fit in MARL's centralized training, we let all controllable agents make predictions based on their observations and calculate their loss, and use the mean of their loss as the final loss.

**Full information agent model (FIAM) (Papoudakis et al., 2021a).** This baseline is a variant of LIAM by replacing the input trajectory of local observations $\tau_t = \{o_{0:t}\}$ with the trajectory of global states $\tau_t = \{s_{0:t}\}$.

**Online adaptation via inferred teamwork situations (ODITS) (Gu et al., 2022).** Unlike the previous two methods that predict the actual behaviors of teammate agents, ODITS improves zero-shot coordination performance in an end-to-end fashion. It has two variational autoencoder pairs, one global and one local. The global encoder takes in state trajectory $\tau_t = \{s_{0:t}\}$ and outputs the mean and variance of a Gaussian distribution. A vector $z_e$ indicating "global teamwork situation" is then obtained by sampling from it. The global decoder uses $z_e$ to build the parameters $z_h$ of a hyper-network that maps the ad hoc agent's local utility $Q_i$ into global utility $Q_{\text{tot}}$ to approach the global discounted return. The local encoder has a similar structure as the global encoder, except that its input is replaced with local trajectory $\tau_t = \{o_{0:t}\}$. It is updated by maximizing the mutual information of its output $\hat{z}_e$ and the global $z_e$. The local decoder further maps $\hat{z}_e$ into a variable $z_c$ used as the ad hoc agent's input. The whole training is end-to-end by maximizing global return and mutual information of $z_e$ and $\hat{z}_e$. Similar to LIAM, ODITS only considers a single ad hoc agent setting. We modified the loss to the mean of all controllable agents' individual loss to make it fit in MARL.

To ensure fairness in the use of hidden variables, we make CSP and all baselines have the same width for $z_c$ in each environment. Concretely, $|z_c|$ is set to 8 for Overcooked, LBF, and PP, and 64 for Fork.

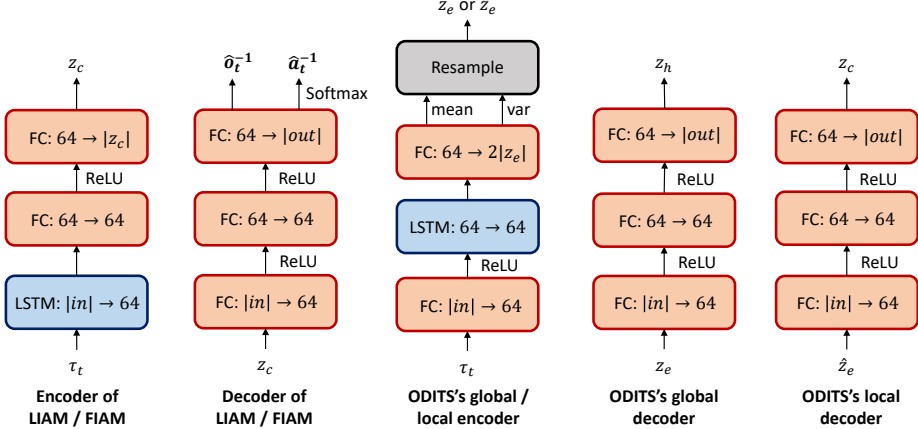

Figure 9: Network architectures of baselines.

# D    MORE EXPERIMENT RESULTS

## D.1    EMBEDDINGS ON EACH ENVIRONMENT

To measure the quality of our scheme embedding, Fig. 10 adopts t-SNE to visualize the embedding of distinguished teams. Each color in four t-SNE sub-figures represents the embedding of distinguished teams. It shows that our scheme embedding can cluster the same coordination scheme and classify the different schemes, which shows a high quality property for team recognition and sub-policy selection.

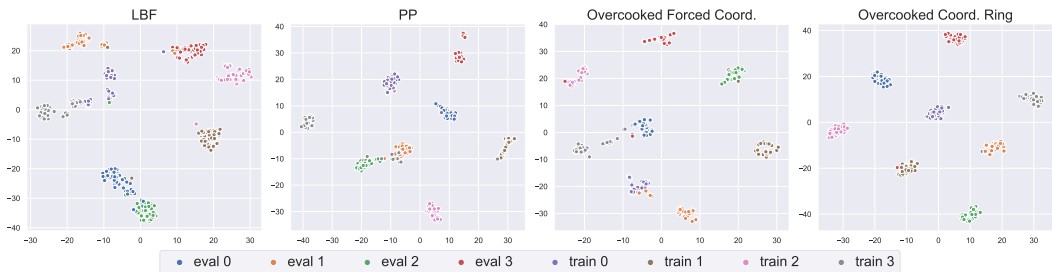

Figure 10: t-SNE of CSP's coordination scheme embeddings in different environments.

## D.2 CROSS-PLAY OF DIVERSE TEAMMATES

To measure the quality and diversity of our SVD population generation in training stage 1. We concatenate all the teams in the training set $\Pi_{\text{train}}$ and the evaluation set $\Pi_{\text{eval}}$ together and make cross play with each other. In LBF, Overcooked, PP, and SMAC, $\Pi_{\text{train}}$ and $\Pi_{\text{eval}}$ are both $4$ while that of SMAC 1c3s5z is $5$. We normalize the score to $[0, 100]$ for each scenario and show them in heatmaps in Fig. 11. In most benchmarks, agents in the population cannot coordinate with others, as the values on the diagonal (i.e. self-play) are much higher. Fork mainly contains two coordination schemes as a comparison. The heatmaps show that our SVD method can generate a diverse population to simulate the underlying coordination schemes.

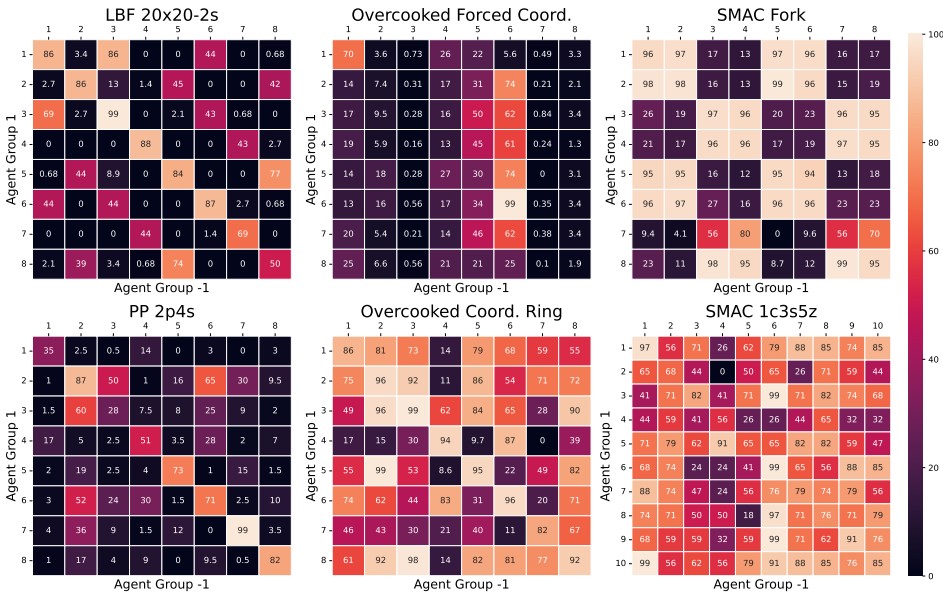

Figure 11: Cross-play performance of $\Pi = \Pi_{\text{train}} \cup \Pi_{\text{eval}}$ in all scenarios.

To further support that $\Pi_{\text{eval}}$ trained with SVD is a better benchmark for testing generalization performance compared to randomly trained teammates, we train another 10 teams independently without SVD for PP and SMAC Fork each. Fig. 12a illustrates their cross-play performance. Compared to the original teams with SVD as shown in Fig. 11, we can find two major phenomenons: (1) It seems that not only the locations at the diagonal have high values, indicating that each team is able to coordinate with some other teams apart from itself. As we have claimed in Sec. 3, coordination performance across different schemes is generally not guaranteed. It is clear that there are multiple teams following the same coordination scheme in the population, which is redundant and makes the population less diverse. (2) We can still see the clear 2-scheme structure for SMAC Fork, but the distribution is biased (7 for one and 3 for the other). An ideal benchmark should be unbiased for all the underlying coordination schemes to best match our goal. SVD encourages different teams to behave as differently as possible, which naturally weakens the potential bias for a particular scheme.

We also tested the performance of CSP and baselines on these populations trained without SVD as shown in Fig. 12b. In this case, CSP still performs better, but the relative advantage is less significant. The result verifies the performance of CSP under random coordination settings, and is also in line with what we claimed above. Since our goal is to train a policy that can coordinate under any scheme, using a more diverse and less redundant $\Pi_{\text{eval}}$ trained with SVD makes sense.

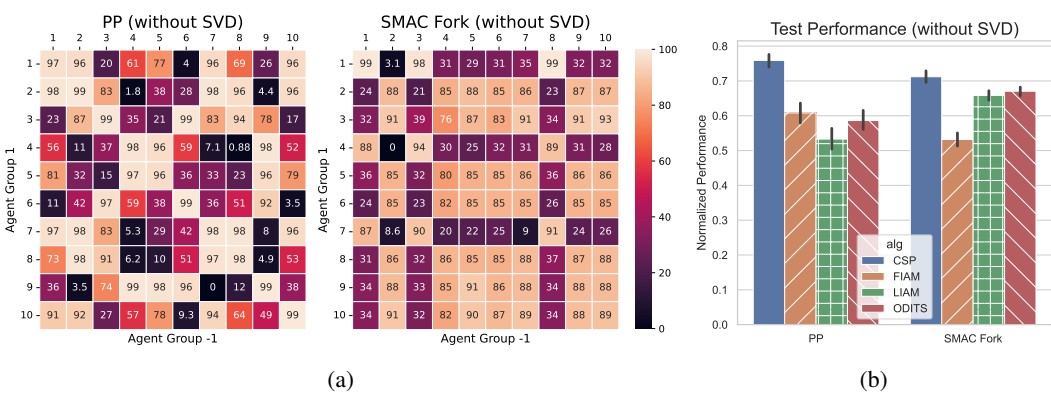

(a)                                                    (b)

Figure 12: (a) The cross-play matrix of populations trained without SVD. (b) Generalization performance of CSP and baselines on the two no SVD populations.

### D.3  EXTRA COST OF CSP COMPARED TO ZERO-SHOT METHODS

As a newly proposed few-shot framework, CSP takes extra environmental interactions and training steps in its additional Stage 2 compared to zero-shot baselines. We draw the learning curves of the self-supervised autoencoder pair $\langle E_c, D_c \rangle$ in Fig. 13 to show how much extra cost is actually required. As can be seen from the plots, the cross-entropy loss between reconstructed teammate actions and the ground truth drops rapidly in the early stage and slowly decreases as training moves on. This phenomenon indicates that the self-supervised learning process in Stage 2 is much more sample-efficient than reinforcement learning. Therefore, although we let Stage 2 to interact with the environment as many timesteps as Stage 3 in our experiments, it actually only requires a small portion of interactions to make a good representation. We will further investigate how to compress this additional cost in future work.

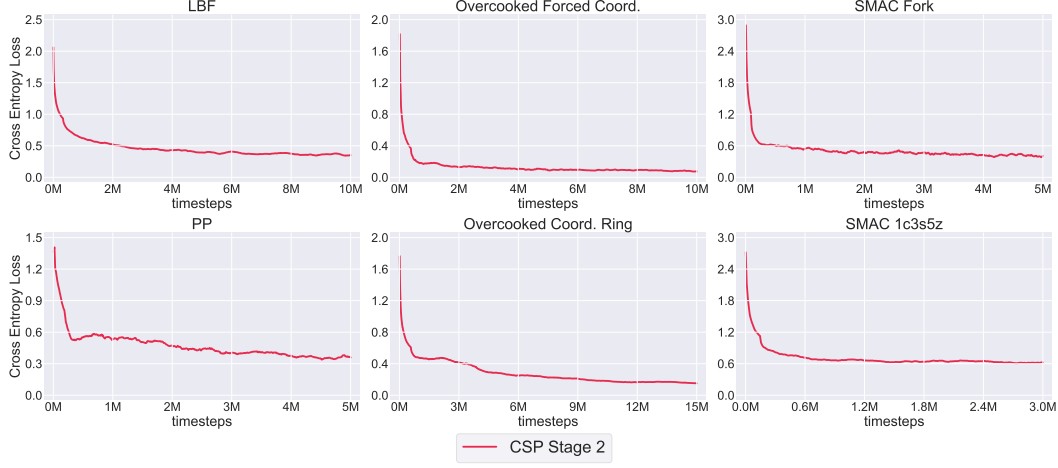

Figure 13: Learning curves of $\langle E_c, D_c \rangle$ in CSP's Stage 2.

