# OpenReview forum: "Coordination Scheme Probing for Generalizable Multi-Agent Reinforcement Learning"
_ICLR.cc/2023/Conference — Submitted to ICLR 2023_

### Official Review · Reviewer_yqCj · 2022-10-19

**Confidence:** 4
**Correctness:** 2
**Technical Novelty And Significance:** 2
**Empirical Novelty And Significance:** 2
**Recommendation:** 3

**Clarity, Quality, Novelty And Reproducibility:**

* Although the paper is well-written, the procedure of the method is complex and unclear, and it is hard to get the main points. A brief algorithm is necessary in the main pages.
* Moreover, many details are not quite clear, e.g., how $\pi_{sp}$ is learned, and MARL($\pi_{sp}$) does not give sufficient information.
* Two-agent settings for the experiments can only be found in Appendix. Such information should be made clear in the main pages.
* For novelty, it seems not quite novel, considering the proposed method is essentially policy representation in two-agent settings.


**Strength And Weaknesses:**

## Strength

* Coordinating with new teammates is an important problem in multi-agent reinforcement learning.
* The proposed method seems reasonable.
* The experiments are performed on a variety of multi-agent tasks.

## Weaknesses
### minor issues
* In problem formulation, the agents under control are denoted by $G^1$ and the teammates are denoted by $G^{-1}$. In Stage 1, do you train the team population of $G^{-1}$ or $G^1 \cup G^{-1}$? I think in multi-agent environments, it is not reasonable to train $G^{-1}$ only, but Stage 1 in Figure 2 is confusing. And I suggest distinguishing $G^1$ and $G^{-1}$ clearly in the method section.

* Interacting with different teammates, the intrinsic reward $L_{pred}$ of scheme probing policy is different. Would the non-stationary reward damage the learning of the scheme probing policy?

* I am curious about whether the team policies in Stage 1 are diverse enough or just collapse to a few modes. The paper claimed that "the number of clusters k will be approximately equal to the number of environmental coordination schemes if $\Pi_{train}$ already covers all possible schemes. " It is important to run experiments with different M (size of $\Pi$) and show the relationship between k and M.

* The intrinsic reward of the scheme probing policy $L_{pred}$ is an important part of CSP. Ablation studies about $L_{pred}$ are necessary to verify the effectiveness.

### major issues
* The problem setting is similar to the meta-learning problem. Thus, meta-learning baselines should be included for comparison, e.g., PEARL [1].
*  The scheme probing module is used to differentiate the embeddings of the trajectories when playing with different teammates. Thus, policy representation methods should also be included as baselines, e.g., [2].
* The proposed method may not be scalable. It seems all coordination schemes should be covered in the training set. Otherwise, the generalization to unseen schemes is not guaranteed. In the current experiments, the settings are essentially two-agent cases (even for SMAC where one agent controls 4 units and another controls 5). What about the case of more than two agents? This should be included as least empirically.

[1] Rakelly et al., Efficient Off-Policy Meta-Reinforcement Learning via Probabilistic Context Variables, ICML 2019.

[2] Grover et al., Learning Policy Representations in Multiagent Systems, ICML 2018.

**Summary Of The Paper:**

Instead of following zero-shot coordination adopted in existing work, the proposed method CSP improves group adaptation using limited interaction data. CSP first trains a set of diverse teammate policies, then train a scheme probing policy to detect the patterns of teammates. By interacting with the teammates, the embeddings of the trajectories of the scheme probing policy encode the information of teammates' patterns (coordination schemes). By clustering the embeddings, CSP obtains the possible coordination schemes. Then CSP trains a meta-policy, which uses each sub-policy to deal with each coordination scheme. In execution, CSP first acts scheme probing policy to interact with the new teammates and then chooses the sub-policy with the closest embedding distance from the scheme probing policy.

**Summary Of The Review:**

More baselines and experiment settings as mentioned above are required. The difference between the proposed method and existing work including policy representation and meta-learning methods should be justified. In its current form, the paper is below the bar of ICLR.

---

> ### Author Response · Authors · 2022-11-12
> **Response to reviewer yqCj (4/4)**
>
> ## Q8 What is the relationship between k and M?
>
> As we have claimed in Sec. 4.3, we think $k$ should grow slower than $M$ and converge to a particular value, which we believe is the size of underlying coordination scheme set $|C|$. At that point, the feasible coordination schemes are already covered in $M$, and no joint-policies with high performance can be further found following a different scheme.
>
> We have conducted an experiment to showcase the relationship between $k$ and $M$. In the SMAC Fork scenario, we created a larger population of size $20$. Then we show the auto-selected $k$ at different $M$ in the table below. The empirical result supports what we claimed in Sec. 4.3 that $k$ seems converging after $M$ reaches a relatively large number, which makes it possible for us to use a relatively smaller $\Pi_{\rm train}$ to achieve a good generalization. It is intuitive that the number of coordination schemes are finite and should not be very large, since each of them corresponds to a unique way of solving the task coordinately, and such high performance solutions are generally limited.
>
> |  M   |  1   |  2   |  3   |  4   |  5   |  10  |  15  |  20  |
> | :--: | :--: | :--: | :--: | :--: | :--: | :--: | :--: | :--: |
> |  k   |  1   |  2   |  3   |  3   |  3   |  3   |  3   |  3   |
>
> ## Q9 About novelty
>
> We agree that some parts of our approach are already proposed in the community (e.g., Policy Representation). But as stated by reviewer d7jH, our contribution is combining these isolated approaches to form an effective and easy-to-use framework to address the practical need for multi-agent generalizable coordination, which has not been deeply considered by the MARL community.
>
> Unlike common meta-learning problems, the distribution of tasks (coordination schemes) is unknown. We utilize diverse population based training to discover these schemes directly from the environment, and further address them with probing, representation and a meta-policy. The whole pipeline is end-to-end, where we only need the access to the environment itself (same assumption to standard RL) to learn a few-shot policy with the ability to coordinate with potentially diverse teammates.

---

> ### Author Response · Authors · 2022-11-12
> **Response to reviewer yqCj (3/4)**
>
> ## Q5 Some details are not clear enough
>
> We would like to thank the reviewer for pointing out these unclear statements. It's our careless mistake that they are not clearly written and may confuse you while reading. We explain them further below, and we have updated these parts in the revised version. If you found any other details that are not clear enough, please feel free to let use know.
>
> **Stage 1:** In fact, we trained the whole team (i.e., $G^1 \cup G^{-1}$) in Stage 1, but only $G^{-1}$ was extracted from it for subsequent use, and $G^1$ was discarded. Such a process is possible thanks to the CTDE framework that we use, where agents make decisions independently during execution. Different agents in a team trained together can be taken out and executed separately after the training is completed.
>
> **Stage 2:** When training $\pi_{\rm sp}$, we treat the teammates in $G^{-1}$ as part of the environment as if only $\pi_{\rm sp}$ is actually interacting. Therefore, we can directly adopt any MARL algorithm to train $\pi_{\rm sp}$ (with the number of agents reduced to $|G^1|$). In this work, we choose to train it with VDN to be consistent with other parts. The same training mechanism is also adopted in Stage 3 to train $\pi_{\rm meta}$ with switching teammates.
>
> **Brief algorithm in main pages**: It is a great idea to include a brief algorithm in the main pages to help readers get a quick understanding. Thank you four your valuable suggestion, and we have added it.
>
> **Full algorithm in appendix**: We have revised the full algorithms in App. A.1 to make them more clear, basically following the above modifications. The control range of $G^1$ and $G^{-1}$ for each stage are marked, especially.
>
> ## Q6 Will the non-stationary reward damage the learning of $\pi_{\rm sp}$?
>
> Yes, we agree that the non-stationary intrinsic reward may cause some damage to the learning of Stage 2 $\pi_{\rm sp}$, but the negative effect should not be significant. In our experiments, the performance of $\pi_{\rm sp}$ is just slightly lower than a regular policy, but the average performance of $\pi_{\rm sp}$ and $\pi_{\rm meta}$ outperforms SOTA baselines as described in Sec. 5.1.
>
> Here we introduce this intrinsic reward mainly to encourage a better exploration. It is worth to mention that curiosity driven [8, 9] and novelty based [10] exploration methods commonly use a similar error term as an intrinsic reward. Their stable performance across a variety of benchmarks are well studied.
>
> [8] Oleksii Zhelo, Jingwei Zhang, Lei Tai, Ming Liu, Wolfram Burgard. Curiosity-driven Exploration for Mapless Navigation with Deep Reinforcement Learning, arXiv 2018.
>
> [9] Jiachen Yang, Igor Borovikov, Hongyuan Zha. Hierarchical Cooperative Multi-Agent Reinforcement Learning with Skill Discovery, AAMAS 2020.
>
> [10] Yuri Burda, Harrison Edwards, Amos J. Storkey, Oleg Klimov. Exploration by random network distillation, ICLR 2019.
>
> ## Q7 Are the populations trained with SVD diverse enough?
>
> Thank you for this reasonable concern, which reflects the inadequacies in our experimental validation. In order to define what is "diverse enough", we need a baseline to use as a reference. In recent works considering generalization in multi-agent scenarios, a common practice is to use a set of teams independently trained with different initializations (random seeds) [5, 11]. We have added an experiment in App. D.2 in the revised version to compare their differences, and validated that populations trained in Stage 1 with SVD are much more diverse than those randomly generated. And the validation set $\Pi_{\rm eval}$ is a more suitable benchmark for testing generalization performance.
>
> The results can be briefly concluded as: (1) Random populations are more likely to hold redundant teams, whose behaviors are alike. Thus, given the same population size, SVD population are more diverse and can cover more schemes. (2) Coordination schemes within random populations are distributed more biased, which is intuitive since we don't know what scheme each independently trained team will acquire. As a comparison, SVD populations are less biased due to its additional behavior diversity regularizer, thus can be used to fairly test generalization performance.
>
> [11] Pengjie Gu, Mengchen Zhao, Jianye Hao, Bo An. Online Ad Hoc Teamwork under Partial Observability, ICLR 2021.

---

> ### Author Response · Authors · 2022-11-12
> **Response to reviewer yqCj (2/4)**
>
> ## Q3 Can CSP solve more agent cases?
>
> CSP actually has the ability to handle multi-agent cases since we adopt VDN [3] (a Centralized Training Decentralized Exestuation (CTDE) [4] method) as the base MARL training framework. In the CTDE setting, the agents are trained centrally but make decisions independently based on local observations during execution, which is the most common practice in current MARL. The $\pi_{\rm sp}$ and $\pi_{\rm meta}$ appearing in the paper are **joint policies**, which are just the joint representations of the **independent policies** of all agents in $G^1$, and can be written as $\pi_{\rm sp}=\\{\pi_{\rm sp,i}\\}, i=1,\dots,|G^{1}|$  and $\pi_{\rm meta}$ likewise.
>
> In the experiments, SMAC Fork and SMAC 1c3s5z are multi-agent scenarios, where both of them have $|G^1|=4$ . During execution, different agents in $G^1$ make their decisions independently based on their own local observations, not by a single centralized policy that directly assigns actions for each agent. And the empirical results on these SMAC scenarios should support the effectiveness of our proposed method in multi-agent cases.
>
> Thank you again for pointing out this very practical concern, and we have revised our paper to make these descriptions clearer. The modifications include: (1) A more detailed description of the joint-policies in Sec. 3.  (2) All $\boldsymbol \pi_i, \boldsymbol \pi_{\rm sp}, \boldsymbol \pi_{\rm meta}$ appear in the paper are in "boldsymbol" form to indicate that they are joint-policies of individual policies.
>
> [3] Peter Sunehag, Guy Lever, Audrunas Gruslys, Wojciech Marian Czarnecki, Vinícius Flores Zambaldi, Max Jaderberg, Marc Lanctot, Nicolas Sonnerat, Joel Z. Leibo, Karl Tuyls, Thore Graepel. Value-decomposition networks for cooperative multi-agent learning based on team reward, AAMAS 2018.
>
> [4] Frans A. Oliehoek, Matthijs T. J. Spaan, Shimon Whiteson, Nikos Vlassis. Exploiting locality of interaction in factored dec-pomdps, AAMAS 2008.
>
> ## Q4 Scalability issue
>
> Thank you for pointing out this issue and we basically agree with your point that if some coordination schemes are not discovered during training, the generalization performance under them can not be guaranteed. We believe this issue arises from the complexity of coordination generalization problem itself. The problem is very different from common multi-task learning or meta-learning problems where different tasks are either previously known or at least drawn from a fixed task distribution. Unknown teammates to coordinate with, however, are not given by an oracle, but can only be discovered by the learning algorithm itself from the environment. Our proposed CSP framework specially optimizes the Soft-Value Diversity (SVD) objective in its Stage 1, which keeps the behaviors of different teams in the population away from each other. Therefore, the population is relatively efficient in covering as many coordination schemes as possible, since their dissimilarity is optimized.
>
> The issue is not unique to CSP, but widely exists in all the similar methods that adopts population based training (PBT) to enhance policy generalization ability [5-7]. They cannot claim an Out-Of-Distribution (OOD) generalization, but instead try to make the population diversified during training. Since CSP is a complete framework aimed at solving the generalization problem directly from the environment in a few-shot manner, we just applied the common practice of diverse PBT in Stage 1 and did not specifically address this existing issue. We agree that scalability issue is a potential limitation for all these methods to be applied to more complex scenarios, but the empirical results support that even we cannot mathematically guarantee the coverage of coordination schemes, CSP still improves generalization by the schemes it already discovered. We will be dedicated to improving scalability in future work. Thank you again.
>
> [5] DJ Strouse, Kevin R. McKee, Matt M. Botvinick, Edward Hughes, Richard Everett. Collaborating with Humans without Human Data, NeurIPS 2021.
>
> [6] Andrei Lupu, Hengyuan Hu, Jakob N. Foerster. Trajectory Diversity for Zero-Shot Coordination, AAMAS 2021.
>
> [7] Rui Zhao, Jinming Song, Haifeng Hu, Yang Gao, Yi Wu, Zhongqian Sun, Yang Wei. Maximum Entropy Population Based Training for Zero-Shot Human-AI Coordination, arXiv 2021.

---

> ### Author Response · Authors · 2022-11-12
> **Response to reviewer yqCj (1/4)**
>
> Thank you very much for carefully reviewing our work and providing constructive suggestions. We have actively incorporated your valuable comments, and our work has improved thanks to your inspiration. We hope our response below can address your concerns. But if there is still something that needs to be addressed, please feel free to let us know.
>
> ## Q1 Meta-learning baseline missing?
>
> The suggested algorithm PEARL[1] has been added to the main experiments in Sec. 5.1, and detailed descriptions of how we implement it has been updated to Appendix C in the revised version.  It is reasonable to consider CSP as a meta-learning method, where different tasks are to coordinate with teammates following various schemes. We fully agree that including a meta-learning baseline can help make the comparison more adequate. Thank you for your constructive advice.
>
> Empirically, PEARL's performance is worse than CSP consistently, and sometimes even worse than zero-shot baselines. We believe this is because PEARL uses context in an end-to-end manner to directly maximize cumulated reward, which is relatively inefficient. The learning process of how to encode such context into a meaningful variable is not as straightforward as CSP, which has a clear goal of representing teammate policies from the context.
>
> [1] Kate Rakelly, Aurick Zhou, Chelsea Finn, Sergey Levine, Deirdre Quillen. Efficient Off-Policy Meta-Reinforcement Learning via Probabilistic Context Variables, ICML 2019.
>
> ## Q2 Policy representation baseline missing?
>
> Apologies - we should have made it clearer about the descriptions of the baselines. We fully agree that policy representation baselines should be included and carefully compared since the key component of CSP is probing and representation. Currently 2 of our baselines are exactly policy representation methods. They are Local Information Agent Modeling (LIAM) and Full Information Agent Modeling (FIAM), both from a recent work [2]. These two methods model teammates' policies with a self-supervised encoder-decoder pair that predicts teammates' current joint observations $\boldsymbol o^{-1}_t$ and joint actions  $\boldsymbol a^{-1}_t$ based on history input. The difference between them lies in the history they can access, where LIAM uses local history $\tau_t=\\{o_\{1:t\} \\}$ and FIAM uses global history $\tau_t=\\{s_\{1:t\}\\}$. The intermediate hidden variable $z$ from the encoder is then considered as the representation of teammate policies and used in downstream RL tasks.
>
> There are two parts of the experiments related to the two baselines. In Sec. 5.1, we compared the generalization performance and found that CSP outperforms both of them, even when FIAM breaks the partial observability constraint. In Sec. 5.2, we illustrated and compared the representation embeddings of CSP against LIAM, and also the accuracy of predicting teammates' joint actions   $\mathbf a^{-1}_t$. Under both of the metrics, CSP did a better job.
>
> Thank you for this valuable point. We have added a subsection of related work about policy representation to Sec. 2 in the revised version, and summarized chosen baselines into three categories to make it more clear:
>
> 1) Meta-learning: PEARL (new)
>
> 2. Policy representation: LIAM, FIAM
>
> 3. Ad Hoc teamwork: PBT, FSP, ODITS
>
> [2] Georgios Papoudakis, Filippos Christianos, Stefano V. Albrecht. Agent modelling under partial observability for deep reinforcement learning, NeurIPS 2021.

---

> ### Author Response · Authors · 2022-12-04
> **Dear Reviewer yqCj, did our response address your questions?**
>
> Dear Reviewer yqCj:
>
> We thank you again for your comments and hope our responses could address your questions. As the response system will end in a week, please let us know if we missed anything. More questions on our paper are always welcomed. If there are no more questions, we will appreciate it if you can kindly raise the score.
>
> Sincerely yours,
>
> Authors of Paper4002

---

### Official Review · Reviewer_d7jH · 2022-10-25

**Confidence:** 4
**Correctness:** 4
**Technical Novelty And Significance:** 3
**Empirical Novelty And Significance:** 3
**Recommendation:** 8

**Clarity, Quality, Novelty And Reproducibility:**

While elements of the ideas have been present in the community, the paper beautifully brings them together to make an effective pipeline. The paper is very clearly written and does a fantastic job of experiments to understand what is going on and what parts are important. The shared codebase was quite easy to follow and hopefully will be released for exploration by the community.

**Strength And Weaknesses:**

**Strengths**

This is a fantasic paper and it's great to see it go all the way towards a practical solution for this problem (albeit computationally expensive). The shared code is also fairly easy to follow and would definitely make it easier to try alternative ideas in this space. The three steps make total sense. While individual one of these may have been tried, this paper bring it together as a strong pipeline for a very practically useful class of problem. SMAC Fork turns out to be surprsingly effective understandable environment as well.

**Weaknesses**

Only unsatisfying part of the solution is the separate step of clustering/grouping which the paper shows was important for effective training. In principle it should be possible to force such clustering by the probing module or be figured out by the conditioned policy.

Given the bilevel framing, it's surprising that PSRO [1] and $\alpha$-PSRO [2] don't even see a mention. Similarly would be useful to mention [3] and [4] when it comes to conditioned policies for multi-agent adaption and their generalization evaluation.

[1] https://arxiv.org/abs/1711.00832

[2] https://openreview.net/forum?id=Bkl5kxrKDr

[3] https://proceedings.mlr.press/v80/grover18a.html

[4] https://www.ifaamas.org/Proceedings/aamas2018/pdfs/p1944.pdf

**Summary Of The Paper:**

The paper focuses on the ad-hoc teaming aspect of coordination. Unlike most previous works it focuses on coordination of different groups of agents rather than a single-agent with different teams. Moreover they show adaption ability in the context of within an episode as well in various multi-agent cooperative scenarios. Their solution has three major components: 1) a bilevel optimization formulation for obtaining a _diverse_ population of teams, 2) a disentangling coordination scheme probing module to classify new teammates with limited episodic data with its own policy optimized for probing actions, and 3) hierarchical policy that chooses the appropriate sub-policy for distinguished coordination scheme to coordinate with other agents.

**Summary Of The Review:**

The paper comes up with a very effective pipeline for solving a super practically important problem of ad-hoc teaming.

---

> ### Author Response · Authors · 2022-11-12
> **Response to reviewer d7jH**
>
> Thank you for carefully reviewing our paper and providing constructive comments and suggestions.  We are very glad that you appreciate our work. Please find our response below.
>
> ## Q1 Why clustering is separated from the probing module?
>
> It is a very good concern. In fact we believe our method stays close to your intuition, which means the clustering process is actually not totally separated from the probing module, but is closely related to its training. Notice that we use $\pi_{\rm sp}$ and $E_c$ to get the embedding set $Z$ of all policies in $\Pi_{\rm train}$ before finally do clustering. This process can be further decomposed into two phases: Firstly use $\pi_{\rm sp}$ interact with all $\pi_i \in \Pi_{\rm train}$ to get a trajectory set $T$. Secondly use $E_c$ to map $T$ into $Z$. When described in this way, the process is very close to the training flow of Stage 2. At the code level, we can optimize the first phase into directly reading the final N trajectories of each team \pi_i\in\Pi_{\rm train} in Stage 2's replay buffer, instead of doing extra cross-plays N times, and all the rest remains unchanged.
>
> To conclude, the clustering process can be viewed as a final step of training the scheme probing module. The module stops further tunning its probing policy $\pi_{\rm sp}$ and autoencoder $\langle E_c,D_c \rangle$, and summarizes its current understanding of $\Pi_{\rm train}$ into $k$ clusters representing the ground truth of coordination schemes. We put this part at the beginning of Stage 3 instead of the end of Stage 2 in the paper mainly because the discovered $k$ is used as a parameter for initializing $\pi_{\rm meta} = \\{\pi_i \mid i=1,\dots k\\}$.
>
> We are not entirely sure that our response matches your concern. So if we missed something, please feel free to discuss it further with us.
>
> ## Q2 More related work
>
> Thank you for pointing us to these high quality works. We have discussed about them in the revised version. Particularly, a new subsection "policy representation" has been added to Related Work, because we have found it is really close to our proposed method and baselines.
>
> PSRO uses a game-theoretic view for MARL and aims at tackling the overfitting problem by learning meta-strategies and reduces complexity with deep cognitive hierarchies. $\alpha$-PSRO further utilizes $\alpha$-Rank as an alternative to Nash solver to avoid equilibrium selection issues and improve efficiency.
>
> Several recent policy representation methods are now added to Related Work alongside your suggested ones. They are mostly unsupervised (self-supervised) methods, aiming at using the representation to anticipate teammate behaviors or stabilize the learning of a fixed team.
>
> ## Extra
>
> We are glad that you mentioned our codebase especially and appreciate its value. It will certainly be released and we intend to keep developing it to improve readability, efficiency and expandability. We sincerely hope our newly proposed framework could draw on the attention of Multi-Agent community to the practical need of generalizable coordination, and the corresponding codebase could make it easier for everyone interested in this problem to get started.

---

> > ### Comment · Reviewer_d7jH · 2022-11-24
> > **Response acknowledgement**
> >
> > Thanks for the update. I agree that the categorizing the last step from stage 2 vs first step of stage 3 is up to us, but what I meant was that figuring out the $k$ clusters directly (maybe via some sort of continuous refinement or constrained objective) would be interesting. I don't think that this paper needs to solve that problem. Possibly overparametrization is even helpful here.

---

### Official Review · Reviewer_27TM · 2022-11-03

**Confidence:** 3
**Correctness:** 3
**Technical Novelty And Significance:** 3
**Empirical Novelty And Significance:** 2
**Recommendation:** 6

**Clarity, Quality, Novelty And Reproducibility:**

Quality:

The paper is generally of a high quality. The related works is sufficiently comprehensive. The paper runs thorough experiments on six scenarios across four different environments, and compares against five state of the art baselines.

Clarity:

The paper is well written and organized, with few typos.

Originality:

The paper combines several existing methods in a novel way to address an under-explored area of research. Furthermore, it introduces some new methods and objectives, namely the Soft-Value Diversity objective and the team-dynamics reconstruction method.



**Strength And Weaknesses:**

Strengths:
* The paper addresses an important question of team-to-team coordination and fast adaptation of the policy.
* The method generally matches or exceeds state of the art on all tasks.
* The paper is generally well written and polished.

Weaknesses:
* This paper seems to be solving a "simple" problem - finding a coordination policy for a particular team - by first solving the much harder problem of finding all possible coordination policies. It's not clear that the SVD objective presented in this paper would be effective in uncovering enough diversity in policies to coordinate well with a general team, as the partner agents are all drawn from the same (evaluation) population and trained with the same method, while prior works have shown that the choice of algorithm can result in substantially different zero-shot coordination ability. It would be interesting to see how the method performs when drawing teams from different populations or partially-trained populations. I would also like to see an ablation of the SVD outer objective during training.

* While there are six different, coordination tasks explored in this paper, most based on prior work, they range of coordination strategies is relatively limited. In the Fork environment for example, there are essentially only two choices strategies, up or down, while in Overcooked Ring the choices are clockwise or counterclockwise. On the opposite extreme, there is no particular strategy to learn for 1c3s5z, to the best of my knowledge. I would be interested in seeing some environments with less clearly defined coordination strategies, such as Hanabi.

Questions:
* Are baselines trained and evaluated on the same populations, trained with the same SVD objective?
* Is the algorithm used QMIX (Section 4.1) or VDN (Section 5)?

Minor:
* Section 3/Problem Formalization - "we" -> "We"
* Section 4/ Meta-Policy Learning - "reset" -> "rest"
* Appendix C/PBT - "tixing" -> "fixing"(?)

**Summary Of The Paper:**

This paper aims to address two weaknesses in current multi-agent coordination methods - they tend to focus on a single agent's ability to coordinate with teams in a zero-shot setting, which ignores team to team coordination and limits the ability to adapt to teammates. The paper addresses these shortcomings using a three step framework:
1) Learn coordination schemes from a diverse set of trained teams by maximizing a Soft-Value Diversity objective, or the variance of softmax normalized value estimates for all policies in a population.
2) Train a module to embed information about teammates' policies using an action autoencoder. The data for this module is collected from one episode of team interaction using a separate probing policy, trained to maximize reconstruction loss of the autoencoder, effectively searching for uncertain states.
3) Classify the teammate embedding within coordination strategies using k-means and learn a meta-policy to select among coordination schemes during evaluation.

**Summary Of The Review:**

Overall, this paper proposes an interesting but fairly complex method. While there are some weaknesses, the thorough experiments and results provide evidence for the effectiveness of the method.

---

> ### Author Response · Authors · 2022-11-12
> **Response to reviewer 27TM (2/2)**
>
> Additionally, the "partially trained" populations as you mentioned are in general not considered in our current setting, since we want teammates to be diverse but still with high quality. In many complex scenarios, even a few less qualified teammates will lead to an overall failure of the task. We think this is an interesting and practical point to consider coordination with teammates with different qualities, and we will consider this in our future work.
>
> [1] Zhenggang Tang, Chao Yu, Boyuan Chen, Huazhe Xu, Xiaolong Wang, Fei Fang, Simon Shaolei Du, Yu Wang, Yi Wu. Discovering Diverse Multi-Agent Strategic Behavior via Reward Randomization, ICLR 2021.
>
> [2] Andrei Lupu, Hengyuan Hu, Jakob N. Foerster. Trajectory Diversity for Zero-Shot Coordination, AAMAS 2021.
>
> ## Q5 Scenario choices
>
> Inspired by your comments, we have made a detailed classification of our scenarios. Thank you for having a deep understanding of the scenarios we choose and giving constructive suggestions. We agree that Overcooked Coordination Ring and SMAC Fork may only have limited coordination schemes.
>
> 1. Have limited and human interpretable coordination schemes: Overcooked 2 scenarios, SMAC Fork.
> 2. Have some coordination schemes, but maybe not easily interpretable: LBF, PP.
> 3. Very restricted optimal strategies, potentially only has a single coordination scheme: SMAC 1c3s5z.
>
> The first type of scenarios are widely used in previous works regarding generalization, especially Overcooked is the most common benchmark for recent Ad Hoc Teamwork papers. We choose them here to be easily comparable to previous methods. Their human interpretability further makes us possible to analyze the behavior of the strategy in depth and thus to verify the effectiveness of various parts of our proposed method. The second type of scenarios are less human interpretable, and the boundaries between different coordination schemes seem less clear. We select them here to show whether the proposed CSP method can mine some coordination schemes that are hard to interpret and make use of them end-to-end. And the comparison against various baselines verifies the effectiveness. The third type of scenarios are chosen to show that the performance of CSP is not worse than other methods even if coordination schemes may not distinct vastly, testifying the safety of applying CSP to arbitrary multi-agent scenarios without the need of priori knowledge.
>
> We would like to thank the reviewer again for pointing us to this very interesting and human playable environment Hanabi. But we believe that the second type of scenarios (i.e., LBF and PP) can reflect similar features as Hanabi to some extend. We will carefully consider this environment in our future work when we dig into human-AI interactions.

---

> ### Author Response · Authors · 2022-11-12
> **Response to reviewer 27TM (1/2)**
>
> Thank you for carefully reviewing our work and providing constructive suggestions. We have actively incorporated your valuable comments to improve our work. We hope our response below can address your concerns.
>
> ## Q1 Are baselines trained and evaluated on the same populations?
>
> Yes. During the training of CSP and all the baselines, $G^{-1}$ is controlled by policies periodically switching within the same $\Pi_{\rm train}$. And the generalization performance is evaluated by the mean of coordination with all policies in the same $\Pi_{\rm eval}$. These two populations $\Pi_{\rm train}, \Pi_{\rm eval}$ are both trained with SVD as described in Stage 1 for each scenario.
>
> ## Q2 Is the algorithm used QMIX (Section 4.1) or VDN (Section 5)?
>
> We clarify that VDN is used for Stage 1 to be consistent with other stages. The expression in Sec. 4.1 has been changed from QMIX to VDN in the revised version. Thank you for pointing this out.
>
> You must have noticed that the choices here are actually open because we only care about how to obtain the sets of diverse policies $\Pi_{\rm train}, \Pi_{\rm eval}$ and not how to jointly train them. You may choose any other MARL methods to replace VDN here, as long as they follow the same Centralized Training Decentralized Execution (CTDE) assumption.
>
> ## Q3 Are we using harder problem to solve simpler one?
>
> The problem we consider is actually not "easy" for the following reasons: (1) Our aim is to coordinate with any reasonable teammates at test time, instead of a particular type of teammates. This makes the problem open and dynamic. (2) We assume no access to the teammate sets and should directly generalize to those we haven't coordinated with before, which is in line with real-world applications. Such limitation makes us not able to directly use meta-learning or multi-task learning methods since they require task distribution. Instead, we have to discover these possible coordination schemes from scratch from the environment by the algorithm itself.
>
> Due to the above reasons, we think firstly finding as many possible coordination schemes as possible makes total sense. We do not need to have a full coverage of schemes to achieve generalization. Empirical results show that the discovered schemes already make a difference, and the way we utilize the diverse training population is more efficient compared to baselines.
>
> ## Q4 Are populations trained with SVD diverse enough?
>
> We have added an experiment in App. D.2 in the revised version to better clarify the diversity acquired through additional SVD compared to vanilla randomly initialized populations. Moreover, generalization performance on random populations of CSP and baselines are also shown to validate the effectiveness of CSP and show that $\Pi_{\rm eval}$ is a suitable benchmark. Thank you for pointing out this issue. We fully understand your concern that $\Pi_{\rm eval}$ is generated in the same way as $\Pi_{\rm train}$, which could introduce bias into evaluation.
>
> The results can be briefly concluded as: (1) Random populations are more likely to hold redundant teams, whose behaviors are alike. Thus, given the same population size, SVD population are more diverse and can cover more schemes. (2) Coordination schemes within random populations are distributed more biased, which is intuitive since we don't know what scheme each independently trained team will acquire. As a comparison, SVD populations are less biased due to its additional behavior diversity regularizer, thus can be used to fairly test generalization performance.
>
> We also tested the performance of CSP and baselines on these populations trained without SVD as shown. In this case, CSP still performs better, but the relative advantage is less significant. The result verifies the performance of CSP under random coordination settings.
>
> Our main purpose of this work is to present an overall few-shot framework for discovering multiple coordination schemes and adapt to them effectively. Thus using SVD in Stage 1 to optimize population diversity explicitly makes sense intuitively and are also verified by the experiments above. It is worth mentioning that other methods dedicated to learn diverse multi-agent policies, such as [1, 2], are also compatible with our framework. We hope this compatibility will facilitate further extensions.
>
> (To be continued)

---

> ### Author Response · Authors · 2022-12-04
> **Dear Reviewer 27TM, did our response address your questions?**
>
> Dear Reviewer 27TM:
>
> We thank you again for your comments and hope our responses could address your questions. As the response system will end in a week, please let us know if we missed anything. More questions on our paper are always welcomed. If there are no more questions, we will appreciate it if you can kindly raise the score.
>
> Sincerely yours,
>
> Authors of Paper4002

---

### Official Review · Reviewer_td12 · 2022-11-06

**Confidence:** 3
**Correctness:** 3
**Technical Novelty And Significance:** 3
**Empirical Novelty And Significance:** 2
**Recommendation:** 5

**Clarity, Quality, Novelty And Reproducibility:**

Being in a niche domain, the paper is understandably hard to write and explain. Thus, it was tough to read and understand properly. So, its clarity was subpar.

The method and experimental setup seem to be good quality and novel. Especially, all the necessary ablations I could think of are present.

The code is provided, so it should be reproducible.

**Strength And Weaknesses:**

## Strengths
- The problem of group-to-group coordination requiring generalization to new teammates is interesting and seems novel.
- The idea of separately probing a newly introduced team makes sense, as a probing policy can explore the parts of the state which bring out coordination schemes. Then, the resultant trajectory would be a good representation of what the team can do in different scenarios. Thus, while solving the task, the policy can look into the potential "future" coordination behavior (assumed to be deterministic though) through the trajectory representation. This eliminates the non-Markovian nature of the problem originating due to the unknown team of agents.
- On careful inspection and understanding, the method and the different proposed stages make sense as a solution to the proposed problem.
- The ablation study is crucial and tests the different variations of the method, albeit in 2 environments.

## Weaknesses
- **Paper writing**:
    + Overall, the paper was hard to follow as someone unfamiliar with the details of ad hoc teamwork. Ideally, the paper could have been written in a way that anyone with a working knowledge of RL and some Multi-agent RL should be able to follow the build-up of the paper. But there were many specialized terms (ad hoc, coordination scheme, etc) introduced without enough details, which made it hard to follow the flow of writing. It took me 2 full reads to even understand what the paper was trying to convey.
    + It is difficult for a reader to distinguish the problem and the solution in the paper. For example, "coordination scheme" is used sometimes as part of the problem formulation and sometimes as part of the approach (e.g. the unsupervised discovered latents are coordination schemes).

- **Unclear definitions**
    + *Coordination scheme definition*: Section 3 just defines coordination schemes as "different ways to solve the task". This is very open-ended and not a concrete definition, which makes it hard to understand the central point of the paper. It leaves many unanswered questions, such as:
        * What is a coordination scheme $c \in C$ exactly? Does it fully define all the sets of optimal behaviors for each agent in the task?
        * Why is the set of coordination schemes C discrete?
        * What does it mean for a policy $\pi^c$ to be derived by a scheme $c \in C$? Is this defined for deterministic policies only?
    + *Problem Formulation*: The formulation of cumulative reward is not explained well. For instance, what does $(\pi^c)^{-1}$ come from and what does it mean? What is the surrogate that is being replaced here?
    + *Assumptions*: The method just lists down the process that CSP follows, but does not first list down the assumptions that the problem setup makes. This makes it hard to understand what an ideal baseline would be. For instance,
        * Why is it assumed that the training can be split into such different stages with full control over the teams that can be sampled and tested?
        * Are the agents at test time also new or only the team compositions are new and expect generalization?
        * Why can't the coordination schemes be different from the training clusters? What if a new coordination scheme is required at test time that none of the clustered policies actually solve well directly, but interpolation is better (i.e. coordination schemes lie on a continuous plane)? In this case, in fact, the POP(z) or CSP(z) ablations should surely work better?

- **Empirical Evaluation**
    + The first (generating diverse team populations and training MARL) and second stages (learning scheme probing policy) of CSP require interaction with the environment. Are these extra environment steps accounted for while plotting the results in Figure 3? It seems from the description in Section 5 that the baselines PBT, FCP, and ODITS do not require Stage 2 interactions; and LIAM, and FIAM do not require Stage 1 or 2 interactions. If that is correct then for a fair comparison of learning efficiency, Figure 3 should account for the extra steps taken by CSP and baselines in stages 1 and 2. I am also curious how many timesteps are currently being dedicated to these stages for the 6 scenarios.
    + While I appreciate the ablation study, it is a bit little unsettling that the POP(z) ablation suffers so much. It could be that the current set of environments and coordination schemes are too simple and discrete that they can be modeled by separate sub-policies. However, for more complex and realistic environments, intuitively the context-based approach, i.e., POP(z) or CSP(z) should be better, as they can model the continuity in coordination schemes and interpolate in its latent representations accordingly. I am curious as to what the authors' thoughts are on this.

## Clarifications
- The introduction mentions: "The information for identifying different teams is collected by the same policy that aims to maximize coordination performance" will suffer from the exploration-exploitation dilemma. Is this validated by comparison to any baseline?


**Summary Of The Paper:**

This paper addresses the problem of group-to-group coordination in multi-agent scenarios, where agents need to coordinate with newly arrived teammates. The proposed CSP approach (1) simulates diverse teams, (2) autoencodes teams' behavior trajectories, and (3) clusters trajectory representations into groups, each of which learns a separate sub-policy. Finally, given a new team to coordinate with, its behavior trajectory is observed and assigned the closest cluster and sub-policy. On four multi-agent cooperative environments, the paper demonstrates zero-shot coordination generalization to unknown teams.

**Summary Of The Review:**

The paper has a good amount of strengths but lacks several details, which limit its understandability, and consequently, a proper evaluation is hard. The empirical evaluation does not yet seem to incorporate the potentially special assumptions made by the paper. If these issues are addressed or clarified, I would be happy to engage in a discussion and raise my score.

---

> ### Author Response · Authors · 2022-11-12
> **Response to reviewer td12 (3/3)**
>
> ## Q6 Overload of Ad Hoc terms affects reading
>
> We are sorry that we may have introduced too many terms without clear definition, which affects your quick understanding. And thank you very much for carefully reading through our paper more than 2 times and giving detailed comments. We fully agree that this work should be written in a form that anyone with some MARL knowledge can easily understand what problem we are trying to address and what our proposed methods are.
>
> In order to express our work more clearly, we have made the following changes in our revised version: (1) We give a more formal definition to **coordination scheme** and modified problem formalization accordingly, as described in Q1. (2) We have streamlined the use of Ad Hoc terms in the Introduction and replaced them with more intuitive and easy-to-understand words. (3) A lot of unclear and ambiguous expressions through the paper have been modified.
>
> Thank you again for carefully reviewing our work and providing constructive suggestions.

---

> ### Author Response · Authors · 2022-11-12
> **Response to reviewer td12 (2/3)**
>
> We didn't take Stage 2 timesteps into account for Stage 3 in the current experiment. Since we intend to develop a new few-shot framework to deal with the problem that no previous method can well address, we think it is reasonable to prioritize final performance over sample efficiency at this stage. As shown in the additional results above, the actual timesteps needed for learning a good representation seem much less than we allow them to train right now. And we plan to further compress this part of the time in our future work.
>
> ## Q4 Why context-based ablations perform worse?
>
> We fully understand your concern, since the worse performance of context-based methods (especially CSP(z)) is counter-intuitive. We think there are two major points that make this happen as shown below.
>
> 1. Noisy representation: The probing procedure requires $\pi_{\rm sp}$ actually interacting with teammates in the environment, where both the environment and the teammates could be stochastic. In this case, the probing trajectory $\tau_{\rm sp}$ of the same teammates can be different, resulting in a nosy representation vector $z$. If we directly use this vector to train a context-based policy, then its learning efficiency could be harmed by such noise. This problem is shared between POP(z) and CSP(z). A potential solution for context-based methods to be stable in this case, is to do multiple times of probing, and use the statistical quantities of all representation vectors. Clustering and classification in CSP naturally play the role of denoising, which can stabilize training.
> 2. Scheme collapse: A single learning policy may start training from a neutral state with no preference for any coordination scheme. But during random exploration, it must discover one of the scheme first, not all the schemes at the same time. Once it finds this particular scheme can lead to larger return (than others), it will develop behavior bias and resolve to explore and learn towards this scheme. As we described in Sec. 3, different schemes are generally inconsistent with each other, thus this behavior bias will be self-reinforcing and forbid the policy to explore other coordination schemes that require a larger behavior jump-out to get the same amount of reward. The policy will collapse to this single coordination scheme and have trouble adapting to others. This problem is unique to POP(z), and the potential solution is exactly what we proposed in CSP, using multiple sub-policies to isolate different coordination schemes and prevent them from affecting each other.
>
> We do not completely deny the validity of the context-based approaches. They can have better local generalization ability due to the continuity in context vector expressiveness and potential interpolations. However, as we have assumed that coordination performance within a single scheme can be guaranteed, the local differences between different policies within a scheme is not significant, and a vanilla policy is sufficient to be their best response. On the contrary, to adapt to different coordination schemes, their dissimilarity and incompatibility require a more aggressive adaptation. In this practical case, the structural design of multiple sub-policies makes sense and performs better empirically just as we showed in Sec. 5.4 ablations.
>
> ## Q5 Can exploration-exploitation dilemma under zero-shot settings be validated?
>
> As we have mentioned in Sec. 1, the exploration-exploitation dilemma should be widespread under zero-shot coordination settings, and can be better solved in few-shot settings.
>
> To get high coordination performance with unknown teammates, a policy should adapt to the same scheme the teammates are under. But in order to figure out which scheme they are under, the policy has to explore some risky or potentially low-return states that can help make a distinction. If these two objectives are considered together in a zero-shot manner (i.e., within a single run), then their incomplete compatibility is obvious. As a comparison, few-shot methods like CSP could decouple these two parts, making early interactions dedicated to probing and representing teammates and the final interaction dedicated to maximizing return.
>
> About empirical validation, the zero-shot policy representation methods LIAM and FIAM perform generally worse than CSP, which we believe can be used as a support for our claim.

---

> ### Author Response · Authors · 2022-11-12
> **Response to reviewer td12 (1/3)**
>
> Thank you very much for carefully reviewing our paper and providing constructive comments and suggestions, which has helped improve our work a lot. We have made some updates to the revised version and hope that our response below can address your concerns. We would be happy to engage in a further discussion with you, so if there remains anything that hasn't been well addressed, please feel free to let us know.
>
> ## Q1 What is coordination scheme exactly?
>
> We have further clarify the formalization as below and all the modifications have been updated in Sec. 3 in the revised version. Thank you for pointing out the ambiguity of coordination scheme, and all the other problems related to it. They are highly constructive comments and let us notice that a mathematical formalization of this concept can help make a lot of things more clear.
>
> **Modifications to Coordination Scheme**: We define this term to better describe generalization. Let $\Pi_f$ be the set of all joint policies with high coordination performance. Coordination scheme $C=\\{c_i\\}$ is defined as a partition of $\Pi_f$. Each coordination scheme $c_i$ is a set of joint policies, where $c_i \cap c_j = \emptyset$, if $i \ne j$ and $\bigcup_{c_i\in C} c_i = \Pi_f$. We assume that the coordination performance can be guaranteed if all the agents are in the same coordination scheme, even if they have minor differences. Otherwise, no such guarantee exists generally. Intuitively, $C$ is determined by the coordination task itself, different elements in which reflects different unique high-level joint behaviors.
>
> **Modifications to problem formalization**: In the objective, the expectations are modified accordingly. And then we use $\pi^{-1}\in \Pi_{\rm train}$ as a surrogate of $\pi^{-1}\in c, c\in C$, where $\Pi_{\rm train}$ is a diverse set we will create in Stage 1.
>
> **Modifications to other parts**: Since coordination scheme currently describes a set, we no longer use words like "represent coordination scheme" in Stage 2, but directly say we are representing team policies. We think this modification can make the meaning of coordination scheme unique (as defined above). In Stage 3, we treat each cluster of team policies as a discovered coordination scheme. The first phase of Stage 3 is then renamed from "Scheme Grouping" to "Scheme Discovery".
>
> ## Q2 What are the assumptions?
>
> Please find our clarified assumptions below. All of them have been updated in Sec. 3 in our revised version. Thank you for pointing them out.
>
> About cooperative task
>
> - **Control range setting**: We assume that we can control all the friendly agents in the Dec-POMDP during training, which is the same as common MARL approaches. But at test time, only $G^1$ is under our control, and $G^{-1}$ is the set of uncontrollable teammates that $G^1$ should adapt to.
>
> About coordination scheme
>
> - **Compatibility guarantee within one scheme**: Just as described in Q1, we assume that the coordination performance can be guaranteed if all the agents are in the same coordination scheme, even if they have minor differences. Otherwise, no such guarantee exists generally.
>
> About problem formalization
>
> - **Teammate scheme consistency**: Since we only assume team performance can be guaranteed within one scheme, teammates in $G^{-1}$ are naturally assumed to belong to the same scheme. We further assume that they do not have adaptation ability during testing, which means $G^{-1}$ will stick to a single scheme no matter what $G^1$ behaves.
>
> ## Q3 What is the extra cost of CSP?
>
> The extra cost of CSP is its Stage 2, and we show below that the cost is acceptable. Firstly we want to clarify that $\pi_{\rm train}$ is used for **all the baselines** including LIAM and FIAM as a basic domain randomization (we have shown in ablations that self-play performs poorly). In this case, the exact extra cost of CSP is its additional Stage 2. In our main experiments, Stage 2 are allowed to train as many timesteps as Stage 3, but we find that learning a good representation for team policies actually requires much less data.
>
> (To be continued)

---

> ### Author Response · Authors · 2022-12-04
> **Dear Reviewer td12, did our response address your questions?**
>
> Dear Reviewer td12:
>
> We thank you again for your comments and hope our responses could address your questions. As the response system will end in a week, please let us know if we missed anything. More questions on our paper are always welcomed. If there are no more questions, we will appreciate it if you can kindly raise the score.
>
> Sincerely yours,
>
> Authors of Paper4002

---

> ### Comment · Reviewer_td12 · 2022-12-04
> **Main concern that empirical evaluation is unfair to the baselines**
>
> Thank you for your rebuttal and edits to the paper. I think the changes made make the paper easier to follow, and some of my understanding-based questions are addressed now. However, some major concerns are there.
>
> ### Not "few-shot" enough
> I understand that the few-shot framework makes sense and is important for quick adaptation to new agents. Having said that, even after probing the coordination scheme, this paper requires comparable timesteps to baselines for adaptation (3M-15M steps over different environments). Intuitively, if Stage 2 is already learning over diverse agent groups and the coordination scheme is not generalized over, then why does there need to be any retraining of policy in Stage 3? Wouldn't simply identifying the coordination cluster should be enough to reproduce a policy that solves the task zero-shot?
>
> ### Unfair comparison to Baselines
> The current empirical evaluation is not fair to the baselines:
> 1. CSP has an extra cost of Stage 2, which is equal to Stage 3 itself.
> 2. CSP also requires an extra episode of probing every time a new agent team enters.
>
> So, effectively CSP utilizes 3x the amount of timesteps that the baseline methods require.
> - To resolve 1., the baselines must be given an equivalent budget to see if they converge to a similar performance as CSP. In most environments, it seems that the baselines have not converged and still have some hope for improvement (e.g., Overcooked Forced Coord, Overcooked Coord Ring).
> - To resolve 2. the baselines must be given either 2x the number of episodes or their episodes should be 2x longer, as compared to CSP.
>
> ### Minor comment about the context-based ablations
> Now that I understand that the coordination schemes are kept the same during training and testing, it makes sense that the context-based versions are poorer than exact clustering. If there is no generalization needed over the coordination scheme (which is the setup of this paper), then this approach makes sense. But it wouldn't make sense for a generalization over coordination schemes.

---

> > ### Author Response · Authors · 2022-12-06
> > **Response to reviewer td12**
> >
> > Thank you for your careful comment. We further clarify some mentioned issues below and hope they can address your concerns.
> >
> > ## 1. The necessity of Stage 3
> >
> > Although Stage 2 is learning over diverse teams in $\Pi_{\rm train}$, the purpose is to probe and represent them. It is not able for $\pi_{\rm sp}$ to directly generalize well and that is why we need Stage 3. As can be found in ablation study, simple domain randomization is not enough to make the policy coordinate well with all the diverse teams. The learning process of Stage 2 is similar to this ablation but with a modified reward that motivates exploration.
> >
> > In a nutshell, Stage 2 produces a probing module that only "knows what your teammates will do". Stage 3 further utilizes that information to adapt accordingly. We disentangle these two stages, but they are both essential.
> >
> > ## 2. Comparison fairness
> >
> > We agree that CSP has extra training cost and the comparison should be done on the convergence performance. Actually, the training timesteps we set are largely extended compared to previous work for better convergence. From our experience, baselines do not have much potential to further improve. We have added experiments on "Overcooked Forced Coord" and "SMAC Fork" to let baselines train 3x timesteps as they originally can, and the results are shown below. It is worth noticing that at some cases baselines are not able to converge but keep fluctuating with high variance, so we report mean and std for last 10% checkpoints to reflect their performance. In Overcooked, the agents will complete several orders within one episode, so there is a chance to make adaptation zero-shot, which makes some baselines relatively competitive. In SMAC, if agents cannot coordinate well from the beginning, they will be destroyed and have no chance to adapt. Therefore, few-shot method CSP has a clear gap to zero-shot baselines.
> >
> >
> >
> > |                          |       CSP       |    LIAM 3x     |    FIAM 3x     |    ODITS 3x    |     PBT 3x     |     FCP 3x     |
> > | :----------------------: | :-------------: | :------------: | :------------: | :------------: | :------------: | :------------: |
> > | Overcooked Forced Coord. | $39.35\pm 4.24$ | $24.47\pm3.67$ | $31.62\pm8.54$ | $35.74\pm8.78$ | $18.07\pm1.91$ | $27.97\pm1.22$ |
> > |        SMAC Fork         |  $7.54\pm0.06$  | $6.08\pm0.09$  | $6.54\pm0.09$  | $5.94\pm0.81$  | $6.00\pm0.23$  | $6.25\pm0.17$  |
> >
> > Additionally, in human-explainable environments (e.g. SMAC Fork), we found that all the baselines tend to "collapse" to one of the coordination schemes and basically ignores what their teammates do. Thus, their performance will be good if teammates happen to belong to that scheme and poor otherwise. Only CSP has the ability to switch between different schemes and generalize well. We think this difference is fundamental and is due to the multiple sub-policy structure. As time permits, we will update all the main experiments with extended timesteps, which should not change the convergence results much.
> >
> > ## 3. Test setting
> >
> > Actually, we do not hope our proposed method can generalize to strictly out-of-distribution coordination schemes, nor does any of the baselines. But at the same time, we don't limit it to working with only the teammates it has met, because $\Pi_{\rm train}$ and $\Pi_{\rm eval}$ are generated independently. The intuition is that coordination schemes are naturally held by the task itself and are relatively limited. Therefore, diverse policy learning in Stage 1 should discover as many different schemes as possible and the test teammates encountered can be more likely to be in-distribution and are easier to generalize.

---

### Author Response · Authors · 2022-11-12
**General response to reviewers**

We appreciate valuable comments from all reviewers. We have revised the paper carefully according to your suggestions. For the sake of clarity, the revisions have been colored in blue. The modification parts are briefly summarized as follows.

- Related work
  - A new subsection "Policy Representation" and more related MARL methods have been added.
- Problem formalization
  - We have further clarified the definition of "coordination scheme" and the assumptions.
- Experiments
  - We have added more experiments, including a new meta-learning benchmark PEARL, diversity comparison between SVD and random generated population, generalization performance to other population, and CSP's extra time consumption analysis.
- Writing
  - We have revised the typos and other issues mentioned by all the reviewers.
  - We have modified the usage of terms according to the modification of problem formalization for clarity.

We hope that our response has addressed all the questions and concerns. But if we missed anything, please let us know. We are always willing to answer any of your concerns about our work and we are looking forward to the following inspiring discussions.

---

### Decision · Program_Chairs · 2023-01-20

**Decision:**

Reject

**Justification For Why Not Higher Score:**

Outstanding concerns regarding the fairness of comparison to baselines in the paper and significance of the additional compute used by the proposed method

**Justification For Why Not Lower Score:**

N/A

**Metareview: Summary, Strengths And Weaknesses:**

This paper motivates the relatively under studied problem of group-to-group coordination. The proposed solution (CSP) contains 3 key stages into an overall system highly praised and supported by one reviewer, but questioned by others. Most notably because CSP utilizes additional compute during its second stage, making comparisons to other baseline approaches in the original submission inconclusive. Furthermore, preliminary results offered by the authors during the discussion period show that by increasing the training time of baselines some perform similarly to CSP when considering variance (e.g. ODITS 3x on Overcooked.) Further exploration of the relative sample efficiency of CSP and (if necessary) improvements to its sample efficiency would significantly improve the algorithmic contribution.